# 9Å structure of the COPI coat reveals that the Arf1 GTPase occupies two contrasting molecular environments

**Svetlana O Dodonova[1,2†], Patrick Aderhold[3†], Juergen Kopp[3], Iva Ganeva[3], Simone Röhling[3], Wim J H Hagen[1], Irmgard Sinning[3], Felix Wieland[3], John A G Briggs[1,4,5]\***

[1]Structural and Computational Biology Unit, European Molecular Biology Laboratory, Heidelberg, Germany; [2]Molecular Biology Department, Max Planck Institute for Biophysical Chemistry, Göttingen, Germany; [3]Heidelberg University Biochemistry Center, Heidelberg University, Heidelberg, Germany; [4]Cell Biology and Biophysics Unit, European Molecular Biology Laboratory, Heidelberg, Germany; [5]MRC Laboratory of Molecular Biology, Cambridge, United Kingdom

**\*For correspondence:** john. briggs@embl.de

[†]These authors contributed equally to this work

**Competing interests:** The authors declare that no competing interests exist.

**Abstract** COPI coated vesicles mediate trafficking within the Golgi apparatus and between the Golgi and the endoplasmic reticulum. Assembly of a COPI coated vesicle is initiated by the small GTPase Arf1 that recruits the coatomer complex to the membrane, triggering polymerization and budding. The vesicle uncoats before fusion with a target membrane. Coat components are structurally conserved between COPI and clathrin/adaptor proteins. Using cryo-electron tomography and subtomogram averaging, we determined the structure of the COPI coat assembled on membranes in vitro at 9 Å resolution. We also obtained a 2.57 Å resolution crystal structure of $\beta\delta$-COP. By combining these structures we built a molecular model of the coat. We additionally determined the coat structure in the presence of ArfGAP proteins that regulate coat dissociation. We found that Arf1 occupies contrasting molecular environments within the coat, leading us to hypothesize that some Arf1 molecules may regulate vesicle assembly while others regulate coat disassembly.

## Introduction

Coated vesicles mediate transport between intracellular organelles. Coat protein 1 (COPI) coated vesicles function in retrograde trafficking between the Golgi apparatus and the Endoplasmic Reticulum (ER), and between the Golgi cisternae in both retrograde and anterograde directions (*Cosson and Letourneur, 1994*; *Letourneur et al., 1994*; *Orci et al., 1997*). Assembly of a COPI coated vesicle is initiated by recruitment of the small GTPase Arf1 (ADP-ribosylation factor 1) to the membrane, where it is activated by a guanine exchange factor and transitions into a GTP-bound state in which its amphipathic N0 helix is inserted into the membrane. Arf1-GTP recruits the protein complex coatomer from the cytoplasm and anchors it at the membrane. Polymerization of coatomer at the membrane assembles the COPI coat, which recruits cargo molecules and membrane machinery to the assembly site (*Harter and Wieland, 1998*; *Jackson et al., 2012*). Growth of the coat, and the resulting membrane bud, increases until a vesicle is pinched off the donor membrane. GTP hydrolysis by Arf1, activated by ArfGAPs (Arf GTPase Activating Proteins), mediates uncoating of the vesicle which subsequently fuses with its target membrane.

Coatomer (the COPI complex) is a heteroheptameric protein complex, consisting of α, β, β', γ, δ, ε and ζ subunits. Coatomer can be subdivided into an outer-coat and an adaptor subcomplex

(*Lowe and Kreis, 1995*). The outer-coat subcomplex consists of α, β′ and ε subunits. α- and β′-COP each contain two WD40 β-propeller domains followed by an α-solenoid, a structural motif that is shared by other coat proteins such as clathrin and sec13/31 from COPII (*Devos et al., 2004*). The adaptor subcomplex consists of β, γ, δ and ζ subunits and is homologous to the clathrin adaptor AP complexes (*Schledzewski et al., 1999*). Within the adaptor subcomplex, βδ and γζ are structural homologues that are likely to have arisen by gene duplication. Each of the large adaptor subunits β- and γ-COP contains an α-solenoid 'trunk' domain, connected by an unstructured linker region to a small appendage domain. The trunk domains each interact with an Arf1 molecule to anchor the coat to the membrane. In contrast to clathrin and COPII, where outer-coat and adaptor are separate complexes, in COPI they form a single protein complex in the cytoplasm that is recruited to the membrane *en bloc* (*Hara-Kuge et al., 1994*).

Insights into the structure and function of coated vesicles have been obtained by crystallography and cryo-electron microscopy (cryoEM) (*Faini et al., 2012*; *Fotin et al., 2004*; *Lee and Goldberg, 2010*; *Shen et al., 2015*; *Stagg et al., 2006*). We recently determined the architecture of the COPI coat assembled on the vesicle membrane in vitro using cryo-electron tomography (cryoET) and subtomogram averaging (SA) (*Dodonova et al., 2015*). We were able to fit homology models and available crystal structures into our cryoET structure to build a molecular model of the assembled coat. Although the COPI adaptor and outer-coat subcomplexes share multiple structural and functional similarities with the clathrin coat, they appear to be organized very differently. The COPI adaptor and outer-coat subcomplexes both form arch-like arrangements on the membrane and together form a highly interconnected coat lattice: there is no clear division of function between outer-coat and adaptor subcomplexes (for consistency we will nevertheless use this nomenclature here).

Multiple subunits in the assembled COPI coat interact with cargo and membrane machinery. The outer-coat COPI subunits α and β′ interact with dilysine K(X)KXX cargo motifs (*Jackson et al., 2012*; *Ma and Goldberg, 2013*), while the adaptor subunit γ interacts with p23 transmembrane peptides (*Cosson and Letourneur, 1994*; *Harter and Wieland, 1998*). COPI coated vesicles are also involved in the transport of K/HDEL cargoes associated with specific receptors (*Semenza et al., 1990*) that interact with the COPI coat (*Majoul et al., 2001*). The transport of HDEL cargoes is dependent on the presence of an amphipathic helix in δ-COP (*Arakel et al., 2016*).

Disassembly of the coat is facilitated by ArfGAPs that induce GTP hydrolysis in Arf1 and coat disassembly using a highly conserved catalytic zinc-finger domain (*Cukierman et al., 1995*). Two classes, ArfGAP1 and ArfGAP2/3, have been implicated in COPI vesicular transport (*Frigerio et al., 2007*; *Liu et al., 2005*; *Poon et al., 1999*). ArfGAP1 contains membrane-binding ALPS (Amphipathic Lipid Packing Sensor) motifs that allow the protein to be recruited to membranes in the absence of coatomer (*Bigay et al., 2005*; *Weimer et al., 2008*). ArfGAP2 and 3 interact with COPI via their non-catalytic, C-terminal regions (*Frigerio et al., 2007*; *Kliouchnikov et al., 2009*; *Yahara et al., 2006*) and require coatomer for efficient membrane recruitment and activity (*Szafer et al., 2001*). ArfGAP2 but not ArfGAP1 was found to be associated with COPI coated vesicles budded from Golgi membranes in vitro (*Frigerio et al., 2007*). Although the functions of all three ArfGAP proteins overlap (*Saitoh et al., 2009*), ArfGAP2 deletion leads to a much stronger retrograde trafficking phenotype than ArfGAP1 deletion (*Poon et al., 1999*). Why COPI is regulated by multiple and different ArfGAP proteins remains unclear.

## Results and discussion

### X-ray structure of β20-390δ2–150-COP

To understand the structure of the COPI coat, and how its assembly and disassembly are regulated, we wished to generate a complete structural model for the assembled coat at sufficient resolution to precisely position the secondary structure elements of the component proteins. Crystal structures were available for the majority of the components of the COPI coat, but not for the βδ-COP subcomplex. As a first step towards a complete structural model, we expressed and crystallized a complex containing β-COP19-391 and δ-COP1-175 from *C. thermophilum* (Ct), from which we obtained a molecular model of βδ-COP comprising residues 20–390 of β-COP ('trunk region') and residues 2–150 of δ-COP (*Figure 1*, *Table 1*). The overall architecture of βδ-COP is similar to that of the homologous γζ-COP (*Yu et al., 2012*). β-COP(20-390) is a curved HEAT-repeat type α-solenoid consisting

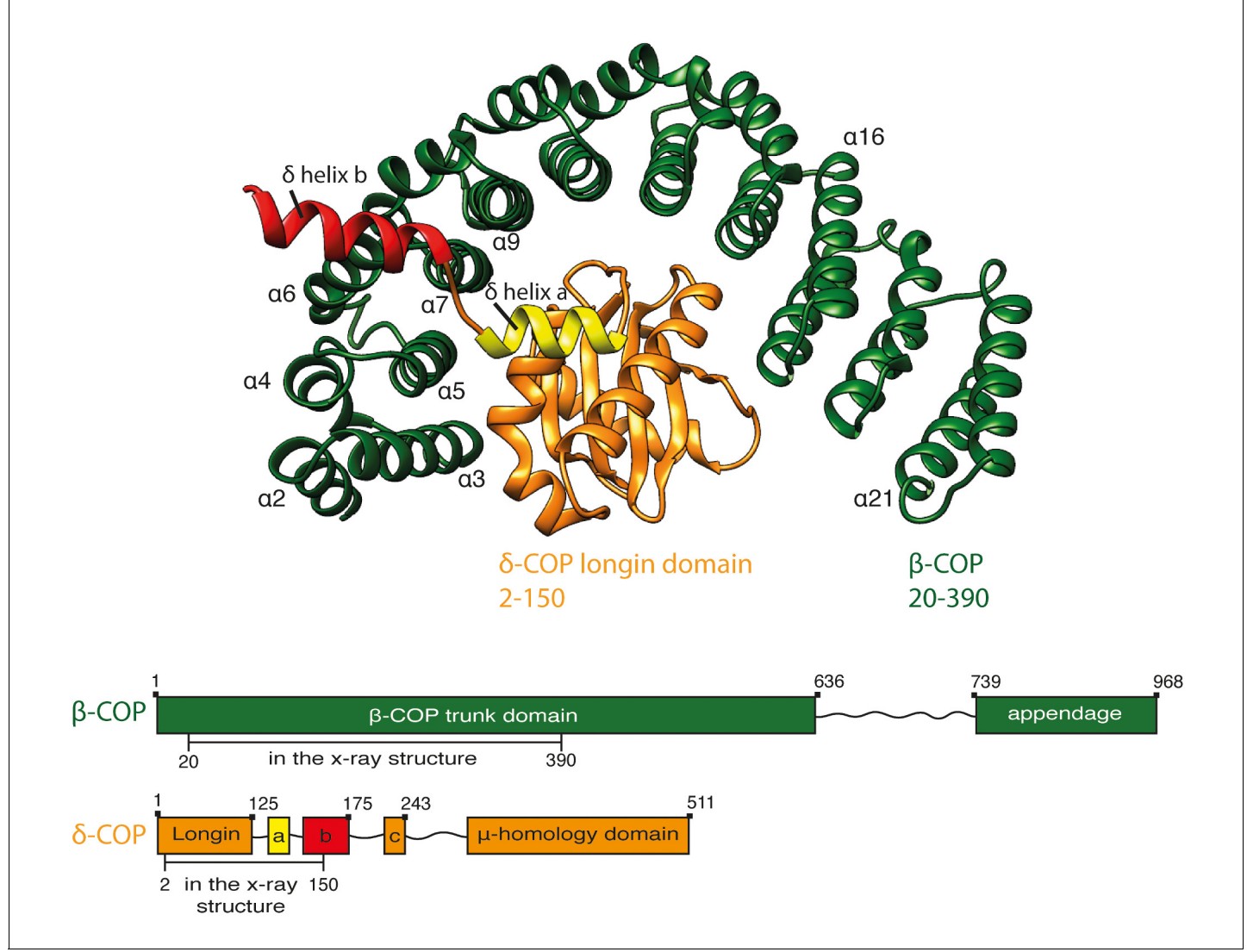

**Figure 1.** Crystal structure of *β*20-390*δ*2–150-COP. A ribbon representation of the structure of Ct*β*20–390 (dark green) Ct*δ*2–150 (orange) as viewed from the vesicle membrane. The α-helices within the α-solenoid of *β*-COP are numbered according to their position in the sequence. δ-COP consists of a longin domain and two downstream helices: helix a and b. Helix b projects away from the α-solenoid of *β*-COP. The schematic at the bottom of the panel shows the subunit domain composition. The subunit regions that are visualised in the x-structure are marked. The *β*-COP sequence numbering includes the N-terminal His-tag and thus is shifted by 15 residues relative to the annotated uniprot-Q9JIF7 *Mus Musculus* sequence. See also *Figure 1—figure supplement 1* and *Table 2*.

The following figure supplement is available for figure 1:

**Figure supplement 1.** The structure of *βδ*-COP.

of 20 consecutive α-helices. The right-handed super-helical arch of *β*-COP wraps around the δ-subunit. The structural model of δ-COP comprises a longin domain followed by a *β*-turn (117-122), and two helices a (127-135) and b (139-150), which are part of the linker region connecting the longin domain and the μ-Homology Domain (MHD). The stretch of residues between helix a and amphipathic helix b traverses the *β*-COP α-solenoid in the vicinity of helix 7. Helix b, including isoleucine residues 143, 146 and 147, projects away from the α-solenoid (*Figure 1—figure supplement 1*) where it would be accessible for binding partners. These isoleucine residues have been shown to be essential for correct trafficking of HDEL motif containing cargo by COPI (*Arakel et al., 2016*).

**Table 1.** Data collection and refinement statistics. Each dataset was collected from one single crystal. Statistics for the highest-resolution shell are shown in parentheses.

| Dataset | SeMet | Native |
|---|---|---|
| Wavelength [Å] | 0.97264 | 0.97625 |
| Resolution range [Å] | 46.40–2.70 (2.83–2.70) | 32.05–2.57 (2.66–2.57) |
| Space group | C 2 2 2$_1$ | C 2 2 2$_1$ |
| Unit cell [Å] | a = 138.22;b = 176.40;c = 62.60 (90°,90°,90°) | a = 137.59;b = 177.47;c = 62.72 (90°,90°,90°) |
| Total reflections | 185546 (6684) | 165162 (19393) |
| Unique reflections | 17882 (1081) | 24909 (2440) |
| Multiplicity | 10.4 (6.2) | 6.6 (6.5) |
| Completeness [%] | 83.2 (39.3) | 99.9 (99.9) |
| Mean I/sigma(I) | 20.8 (1.9) | 15.1 (1.3) |
| R$_{merge}$ | 0.069 (0.819) | 0.068 (1.391) |
| R$_{pim}$ | 0.031 (0.523) | 0.031 (0.644) |
| CC1/2 | 0.999 (0.620) | 0.999 (0.502) |
| # Reflections used in refinement | | 24904 (2439) |
| # Reflections used for R$_{free}$ | | 1228 (119) |
| R$_{work}$ | | 0.1748 (0.2802) |
| R$_{free}$ | | 0.2273 (0.3185) |
| Number of non-hydrogen atoms | | 4213 |
| in Macromolecules | | 4169 |
| Protein residues | | 521 |
| RMSD on bonds [Å] | | 0.009 |
| RMSD on angles [°] | | 1.19 |
| Ramachandran favored [%] | | 97 |
| Ramachandran allowed [%] | | 2.5 |
| Ramachandran outliers [%] | | 0 |
| Average B-factor [Å$^2$] | | 88.27 |
| Macromolecules [Å$^2$] | | 88.42 |
| Solvent [Å$^2$] | | 74.47 |

## CryoET structure of the assembled coat

We prepared COPI coated vesicles in vitro as described previously (*Dodonova et al., 2015*) and vitrified them for cryoET. Tomographic data was collected making use of a direct electron detector and optimized data collection conditions (*Hagen et al., 2017*). Subtomogram averaging was performed essentially as described in the Materials and methods, and the structure was further refined by combining multiple locally-aligned structures. From 1733 vesicles and near-complete buds we obtained a structure of the leaf (representing one coatomer complex with two Arf1 molecules), the asymmetric unit of the COPI coat, at 9.2 Å resolution (*Figure 2*, *Figure 2—figure supplement 1*, *Video 1*). At this resolution, distinct α-helical densities and β-propeller blades are resolved (See also *Figure 3A and B*).

In order to generate a complete pseudo-atomic model of the COPI coat, we performed rigid body fitting of the available crystal structures of COPI components, including our new x-ray structure of βδ-COP, into the structure of the leaf. Next we fitted the homology models of domains for which high-resolution structures are not available (α-COP 327–813, β-COP 410–968, γ-COP 312–549), into the leaf EM map. It is important to note, that the COPI complex is highly conserved among eukaryotes. The sequence identity between the βδ-COP from *Chaetomium thermophilum* used for crystallography and *Mus musculus* used for subtomogram averaging is 53%. The excellent match between

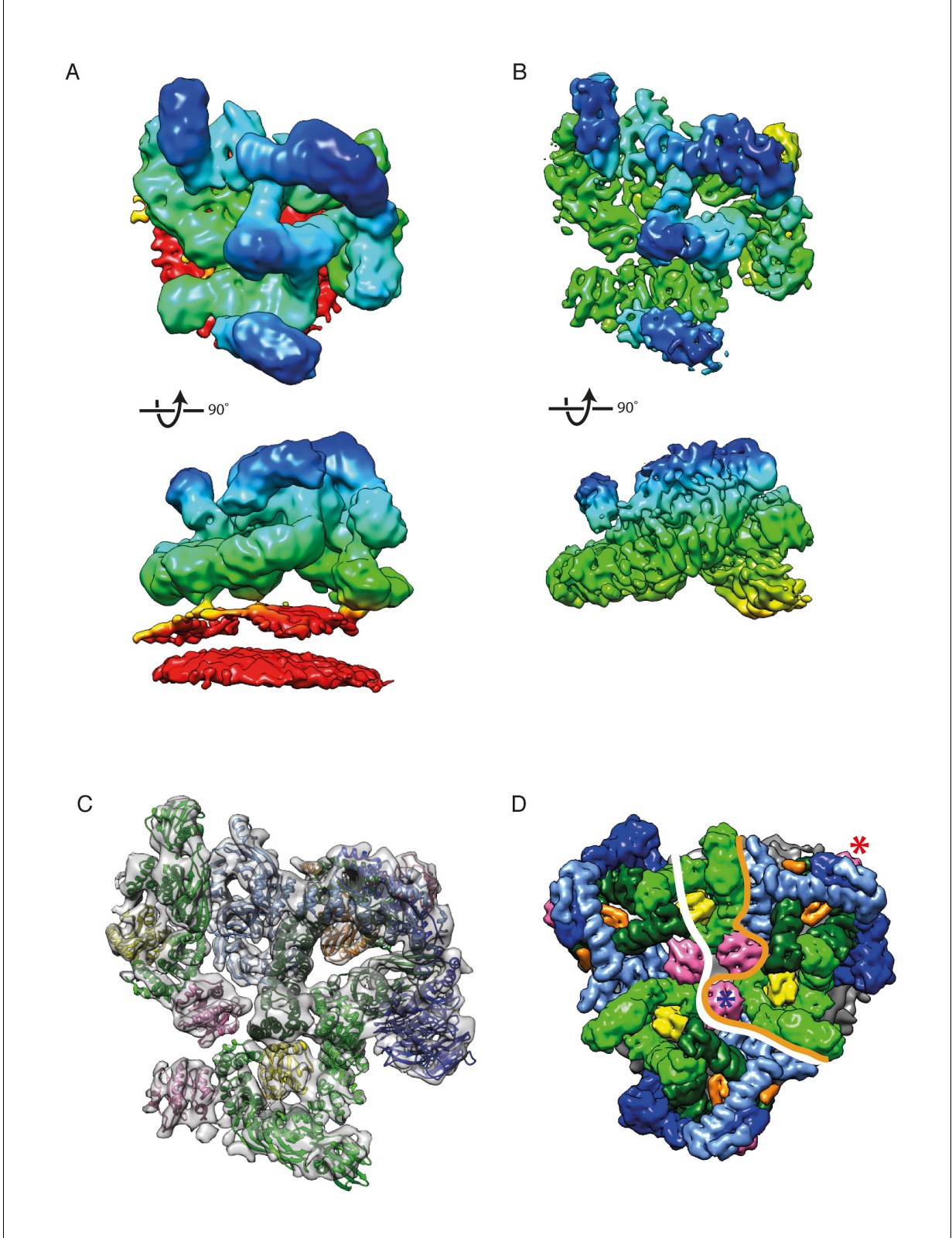

**Figure 2.** The structure of the COPI leaf at 9 Å resolution. (**A**) CryoET reconstruction of the COPI asymmetric unit, the 'leaf' before local alignments. The density is colored according to the distance from the membrane - from red to blue. The displayed structure also contains part of Arf1-γ-ζ-COP from a neighboring leaf to show inter-leaf interactions. The density is displayed at 0.04 isosurface level in order to visualize the membrane. (**B**) CryoET reconstruction of the leaf at 9.2 Å resolution after local alignment. The membrane was masked out during local alignments and upon generation of the

*Figure 2 continued on next page*

*Figure 2 continued*
combined map. Note the definition of the α-helical densities in the structure. (**C**) A structural model of the COPI coat after flexible-fitting of structures and homology models into the cryoET structure. Note, that the C-terminal domain of α-COP, ε-COP, and the δ-COP MHD, are not visualized in the leaf structure, since they compose the inter-triad linkages. Color scheme: cryoET density - grey, Arf1 - pink, γ-COP – light green, β-COP - dark green, ζ-COP - yellow, δ-COP - orange, β'-COP - light blue, α-COP – dark blue. (**D**) The COPI triad. One asymmetric unit, the 'leaf', is outlined with an orange line. The part of the structure displayed in this figure A-C is outlined with the white line. The central Arf1 (γArf1) is marked with a blue asterisk, the peripheral Arf1 (βArf1) with a red asterisk. COPI subunits are displayed as molecular surfaces. See also *Figure 2—figure supplement 1*, *Figure 2—figure supplement 2*, and *Video 1*.
The following figure supplements are available for figure 2:

**Figure supplement 1.** The COPI coat structure at 9 Å or 13 Å resolution.
**Figure supplement 2.** COPI coat linkages.
**Figure supplement 3.** The spatial relationships between components of the coat are illustrated here as an aid to the reader.

the fitted structures and the density further confirmed the structural conservation of the complex as well as the quality and validity of the map (see also *Figure 1—figure supplement 1E*). Flexible fitting (*Trabuco et al., 2008*) was performed to relax the structures and models into the map and generate an initial structural model for the assembled coat. While the core subunits are in positions very similar to those in our previous model (*Dodonova et al., 2015*), there are larger changes in the positions of subunits for which only homology models were available. The improved resolution allowed us to resolve ambiguities in the orientations of appendage domains (*Figure 3*). Most importantly, since individual secondary structure elements are resolved, we were able to identify and assign structural elements that function separately from the folded domains (see below). The higher resolution of the EM map when compared to our previously published study (*Figure 2A–B*, *Figure 2—figure supplement 1C–E*), and the availability of the βδ-COP structure, therefore allowed us to generate a more complete and significantly improved pseudo-atomic model of the coat (*Figure 2C–D*).

## β-COP and γ-COP appendage domains

The γ- and β-COP appendage domains are homologous to the α and β2 ear/appendage domains of the clathrin adaptor AP-2. Both appendage domains are important for viability in yeast (*DeRegis et al., 2008*; *Hoffman et al., 2003*). In our previous model we found that the γ-COP appendage domain is bound to the second β-propeller of β'-COPI, while the β-COP appendage domain is tucked into the center of the leaf between α-COP and γ-COP. Our new structure shows that γ-COP interacts with β'-COP via a highly conserved interface in the base of the γ-COP β-sandwich subdomain (*Figure 3A,B,C*). The platform subdomain of the γ-COP appendage is exposed (*Video 1*) and easily accessible for other binding partners such as ArfGAP (*Watson et al., 2004*). The β2 and α2 appendage domains of AP2 are also thought to function as recruitment hubs for interaction partners, suggesting that these domains are also in an accessible location within the clathrin vesicle (*Praefcke et al., 2004*; *Schmid et al., 2006*). To date, there are no structures available showing the positions of the AP2 appendage domains relative to AP2 or the clathrin coat.

We were previously unable to determine the orientation of the β-COP appendage domain. In the new structure, the β-sandwich and the platform subdomains of the β-COP appendage are clearly visible in the EM map (*Video 1*). The

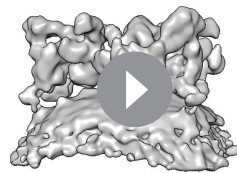

**Video 1.** The COPI coat structure and model.
A tour through the COPI structure highlighting key features of the coat discussed in this study.

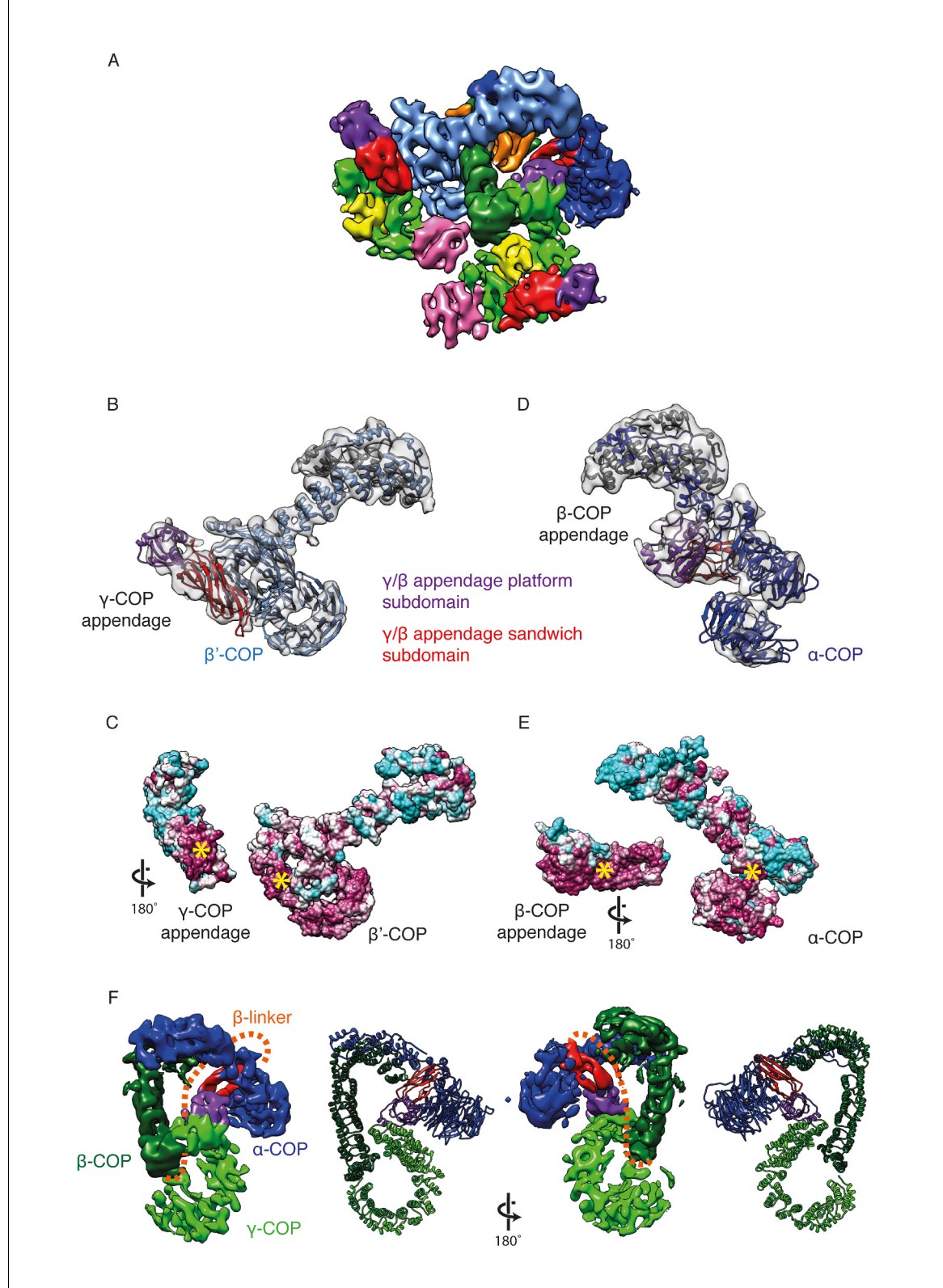

**Figure 3.** γ-COP and β-COP appendage domains within the coat. (**A**) Localization of the γ-COP and β-COP appendages in the COPI leaf. The 9.2 Å EM map of the COPI leaf is colored based on the underlying subunit. Color scheme as in *Figure 1*, additionally appendage sandwich subdomains are red and appendage platform subdomains are purple. (**B**) γ-COP appendage (sandwich subdomain – red, platform subdomain - purple) interacts with the *β'*-COP (light blue). The models are shown within the corresponding part of the COPI EM map (transparent grey isosurface) (**C**) The structure as in *Figure 3 continued on next page*

*Figure 3 continued*

B, opened out to reveal the conserved interaction interfaces of γ-COP and β'-COP (purple – conserved, cyan - variable). The yellow asterisks mark the interaction surfaces. (**D**) β-COP appendage interacts with α-COP (dark blue). (**E**) The structure as in D, opened out to reveal the conserved interaction interfaces of β-COP appendage and α-COP (purple – conserved, cyan - variable). (**F**) The β-COP appendage domain provides the main structural connection between the γ-COP adaptor subunit (light green) and the outer-coat subunit α-COP (blue), where it interacts with both β-propeller domains. Organization of the β-COP subunit: the long β-COP trunk domain (dark green) is connected to the appendage domain (red/purple) by a flexible linker (dashed orange line). We note that the β-COP trunk domain conformation, which is rather straight, is significantly different from the conformations of the β-subunits from homologous AP complexes, which are highly curved. The panel on the right shows the subcomplex from the 'bottom' (from the membrane side) in order to visualize the suggested β-linker path more clearly.

platform interacts with the trunk domain of γ-COP, whereas the β-sandwich forms a conserved interface with α-COP including both α-COP β-propeller domains and part of the α-COP α-solenoid domain (*Figure 3D*). In the clathrin system, the β2-appendage domain of AP2 plays a conceptually similar role, interacting with clathrin and promoting cage formation (*Owen et al., 2000*). The β2-appendage is thought to be linked to the core of AP2 only by its flexible linker. In contrast, the β-COP appendage appears to form the main link between the adaptor and the outer-coat subcomplexes of the COPI coat. The other connection is a rather small interface between the β-COP trunk and β'-COP. We speculate that the position of the β-COP appendage domain would allow it to modulate the conformation of coatomer by fixing the angle between the two β-propeller domains of α-COP and by forming a rigid buttress between the adaptor subunit γ-COP and the outer-coat subunit α-COP (*Figure 3D*, *Video 1*), thereby determining their relative arrangements. Thus, the β-COP appendage appears to act as a keystone in the assembled COPI coat.

## Assignment of extra densities

After fitting known structures and homology models to generate the initial structural model, we identified seven positions in the map where there were substantial regions of electron density not occupied by any of the fitted coatomer domains (see Materials and methods) (*Figure 4A*). We wished to identify the protein components that contribute these extra densities. We first performed secondary structure predictions for all COPI subunits using the Quick2D server (*Jones, 1999*; *Ouali and King, 2000*). We identified putative secondary structure elements in multiple COPI subunits that were outside the regions included in the crystal structures and homology models of the COPI domains. We compared the sequence positions of these elements with the locations of the unoccupied densities, as well as with available cross-linking mass-spectrometry (XL/MS) data, thereby assigning secondary structure elements to unoccupied densities.

The C-terminal region of β'-COP was predicted to contain two additional α-helices, consistent with the presence in the EM density of elongated, unoccupied densities adjacent to the C-terminus of β'-COP at the β'-α interface (*Figure 4A,B* '1'). We speculate that these helices may stabilize or regulate flexibility at the β'-α solenoid-solenoid interface.

We interpret the unoccupied density adjacent to the N-terminal β-propeller of α-COP, in the vicinity of the Arf1 bound to β-COP (βArf1), as being partly contributed by a flexible hydrophobic loop of the β-propeller itself that was absent from the crystal structure of the propeller domain (*Figure 4C* '2') (PDB:4J87, [*Ma and Goldberg, 2013*]). Flexibility of the loop likely results in smearing of the density, and we cannot rule out that this density also contains other protein components (e.g. unstructured linker regions).

Five additional α-helices were predicted C-terminal to the β-trunk domain, consistent with the presence of several unoccupied densities near the C-terminal end of the fitted β-trunk model (*Figure 4A,D* '3' and '4'). Cross-links between these extra helices and the ζ-longin domain support their location in that region (β617- ζ39, β618- ζ39) (*Table 2*). After these additional helices, the 103 amino acid unstructured β-COP linker extends in the sequence until the start of the β-appendage domain. The theoretical fully extended length of the linker is around 370 Å. Within the assembled coat it must span a distance of approximately 100 Å. Several cross-links between the linker, the β-COP appendage domain, β'-COP, and α-COP (*Table 2*, rows 5–15) are consistent with the expected route of the linker through the assembled coat, schematically shown in *Figure 3F*. The small density

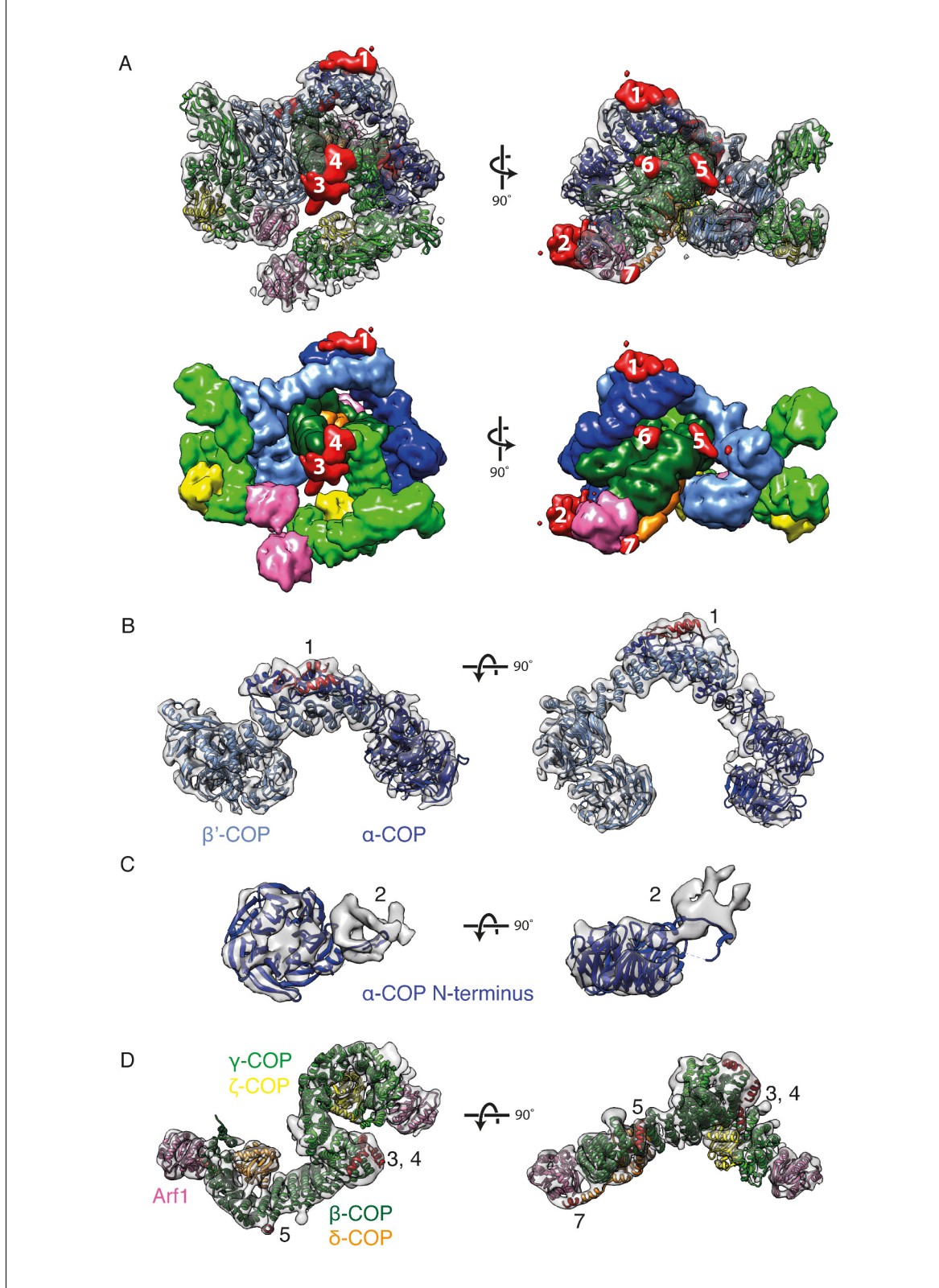

**Figure 4.** New densities identified in the COPI structure. (**A**) Densities within the EM map that are not occupied by the fitted domain structures are shown in red and numbered from 1 to 7 according to their size. The fitted molecules in the lower panel are shown as surfaces. (**B**) The outer-coat subunits of COPI and a newly assigned region. Density 1 is adjacent to the α-COP/β′-COP interface and can be fitted with the α-helices (red) from the C-terminal part of the β′-COP subunit. (**C**) The density near the α-COP N-terminal β-propeller domain. The extra density at least partly originates from
*Figure 4 continued on next page*

*Figure 4 continued*

the mobile loop on the side of the β-propeller domain. (**D**) The adaptor subunits of COPI and three newly assigned regions. Densities 3 and 4 are located near the C-terminus of the β-trunk domain and are fitted with the additional β-COP C-terminal helices. Densities 5 and 7 are located in the vicinity of the δ-COP subunit and are fitted with the helices belonging to that subunit (see also *Figure 5*). See also *Figure 4—figure supplement 1*.

The following source data and figure supplement are available for figure 4:

**Source data 1.** Source data file *Figure 4—figure supplement 1B*.

**Figure supplement 1.** Identification of extra densities in the COPI EM map.

'6' (*Figure 4A*), may represent a short helical stretch in the linker bound to the β-trunk. Density '5' is discussed below.

## δ-COP helices 'a, b and c'

Secondary structure predictions indicate the presence of three putative α-helices (referred to as 'a, b and c') in the linker-region between the δ-COP longin domain and MHD (*Figure 5A*).

Helix a and the first part of helix b are present in our x-ray structure (*Figure 1*). One short and one long extended rod-like density corresponding to these helices are also observed in equivalent positions immediately C-terminal to the fitted δ-COP longin domain in our COPI coat EM density (*Figure 5B*). While no electron density was visible for residues 151–175 of δ-COP in the x-ray structure, our cryoEM map of the COPI coat has clear density corresponding to a longer helix, and we modeled residues 139–165 of the predicted helix b into our density (*Figure 5*).

Helix a interacts with the δ-COP longin and the β-COP trunk domains. The N-terminal part of helix b interacts with the β-COP trunk domain, while its C-terminal part interacts with the peripheral Arf1 molecule (βArf1) in the triad (*Figure 5B*). The very C-terminal end of helix b also contacts the membrane surface (*Figure 5C*). An equivalent helix b is predicted in the sequence of the homologous ζ-COP (*Alisaraie and Rouiller, 2012*), but we saw no equivalent density near ζ-COP in our EM structure.

In an electron microscopy structure of a soluble trimeric assembly of AP-1, Nef, and Arf1 (*Shen et al., 2015*) an interaction was observed between the μ1 subunit of AP-1 (homologous to δ-COP), and Arf1. The interaction made by μ1 is very different to that made by δ-COP, since it involves the MHD of μ1 instead of the longin domain, and a different surface of Arf1.

Several cross-links (δ233-δ263, δ243-δ263, δ233-β'627 and δ241-β'627) (*Table 2*) suggest the approximate location of helix c (*Figure 6—figure supplement 1*), and based on these observations we speculate that density '5' is occupied by this α-helix (*Figure 4A,D* '5'). In this position, helix c could help to coordinate the positioning of the C-terminal δ-COP MHD, which could otherwise be located a long distance from the vesicle: there are ~100 amino acids between helix b and the MHD.

## Interactions between δ-COP and Arf1

Further validation of the interaction between δ-COP helix b and Arf1 is provided by published photo-cross-linking data (*Sun et al., 2007*), which showed a cross-link between Arf1 residue 167 and δ-COP (*Figure 6C*, Arf167 is marked), as well as by mass-spectrometry cross-linking data that showed cross-links between the second longer helix b and Arf1 (δ142-Arf36, δ164-Arf36) (*Table 2*, *Figure 6C*). We prepared δ-COP variants containing photolabile amino acids in position 156 or 159. These variants were expressed as subcomplexes with β-COP(19-391) and recruited to liposomes in an Arf- and GTP-depended manner. Subsequent photo-cross-linking resulted in an 80 kDa product that was confirmed by western blotting to consist of Arf1 and δ-COP (*Figure 6A*). We also prepared an Arf1 variant with two photolabile amino acid derivatives in positions 46 (known to cross-link to β-COP [*Sun et al., 2007*]) and 167. This Arf1 could be photo-cross-linked within vesicles to give a 180 kD product, corresponding to β-COP, δ-COP and Arf1 (*Figure 6B*, and *Figure 6—figure supplement 2*), confirming that one Arf1 molecule interacts with both β- and δ-COP within the same coatomer complex.

To dissect the role of the δ-COP linker-region (δCOP 117–271) in binding to Arf1, we made four δ-COP constructs incorporating different parts of this linker region. These terminate after helix a

**Table 2.** Cross-links from newly assigned COPI domains The mass-spectrometry cross linking data is part of a previously published dataset (***Dodonova et al., 2015***). The distances between lysine pairs for which cross-links were observed were measured for our structural model. If the measured distance was below 35 Å, it satisfied the distance criteria.

| # | Subunit 1 | a.a. | Domain 1 | Subunit 2 | a.a. | Domain 2 | ID score | Distance, nm |
|---|---|---|---|---|---|---|---|---|
| 1 | β' | 856 | C-terminal part | α | 1180 | CTD | 21,25 | — |
| 2 | β' | 871 | C-terminal part | α | 1180 | CTD | 24,46 | — |
| 3 | β | 617 | loop near extra helix | ζ | 39 | core | 27,11 | — |
| 4 | β | 618 | loop near extra helix | ζ | 39 | core | 24,31 | — |
| 5 | β | 666 | linker | β | 891 | appendage | 46,62 | — |
| 6 | β | 671 | linker | β' | 779 | α-solenoid | 32,34 | — |
| 7 | β | 671 | linker | β | 891 | appendage | 47,02 | — |
| 8 | β | 694 | linker | α | 750 | α-solenoid | 30,68 | — |
| 9 | β | 718 | linker | β' | 779 | α-solenoid | 28,42 | — |
| 10 | β | 718 | linker | α | 671 | α-solenoid | 38,19 | — |
| 11 | β | 718 | linker | α | 698 | α-solenoid | 36,71 | — |
| 12 | β | 718 | linker | β | 324 | trunk | 38,14 | — |
| 13 | β | 718 | linker | β | 801 | appendage | 35,16 | — |
| 14 | β | 718 | linker | β | 891 | appendage | 35,1 | — |
| 15 | β | 727 | linker | α | 676 | α-solenoid | 25,12 | — |
| 16 | δ | 142 | extra helix b | Arf1 | 36 | core | 28,0 | 22 |
| 17 | δ | 142 | extra helix b | β | 122 | trunk | 36,07 | 16 |
| 18 | δ | 142 | extra helix b | β' | 69 | 1st β-propeller | 31,54 | 60 |
| 19 | δ | 156 | extra helix b | β | 122 | trunk | 31,37 | 24 |
| 20 | δ | 156 | extra helix b | β | 47 | trunk | 26,7 | 27 |
| 21 | δ | 164 | near extra helix b | β | 122 | trunk | 30,62 | — |
| 22 | δ | 164 | near extra helix b | Arf1 | 36 | core | 33,33 | — |
| 23 | δ | 224 | linker | β' | 615 | α-solenoid | 18,14 | — |
| 24 | δ | 227 | linker | δ | 309 | MHD | 33,02 | — |
| 25 | δ | 231 | linker | δ | 309 | MHD | 33,28 | — |
| 26 | δ | 233 | extra helix c | β' | 627 | α-solenoid | 29,71 | 26 |
| 27 | δ | 233 | extra helix c | δ | 243 | extra helix c | 38,83 | 16 |
| 28 | δ | 233 | extra helix c | δ | 256 | linker | 35,08 | — |
| 29 | δ | 233 | extra helix c | δ | 263 | MHD | 39,82 | 36 |
| 30 | δ | 241 | extra helix c | β' | 627 | α-solenoid | 24,78 | 18 |
| 31 | δ | 243 | extra helix c | δ | 263 | MHD | 24,59 | 23 |
| 32 | δ | 256 | linker | δ | 263 | MHD | 31,39 | — |
| 33 | δ | 256 | linker | δ | 309 | MHD | 33,11 | — |
| 34 | δ | 256 | linker | δ | 322 | MHD | 24,85 | — |

(δ1–137), helix b (δ1–175) or helix c (δ1–243) or code for full-length δ-COP. These proteins were expressed as subcomplexes with β-COP(19-391) and tested for binding to Arf1 in its GMPPNP-loaded state using pulldown assays (***Figure 6D***). The complex containing δ1–137 showed very little GTP dependent Arf1 binding, δ1–175 showed an intermediate level, while δ1–243 bound Arf1 in a GTP dependent manner at the same level as the complex containing full length δ-COP (***Figure 6E***). These results indicate that the interaction between βδ-COP and Arf1-GTP is stabilized by δ-COP helix b and is further stabilized by downstream regions of δ-COP including helix c. These

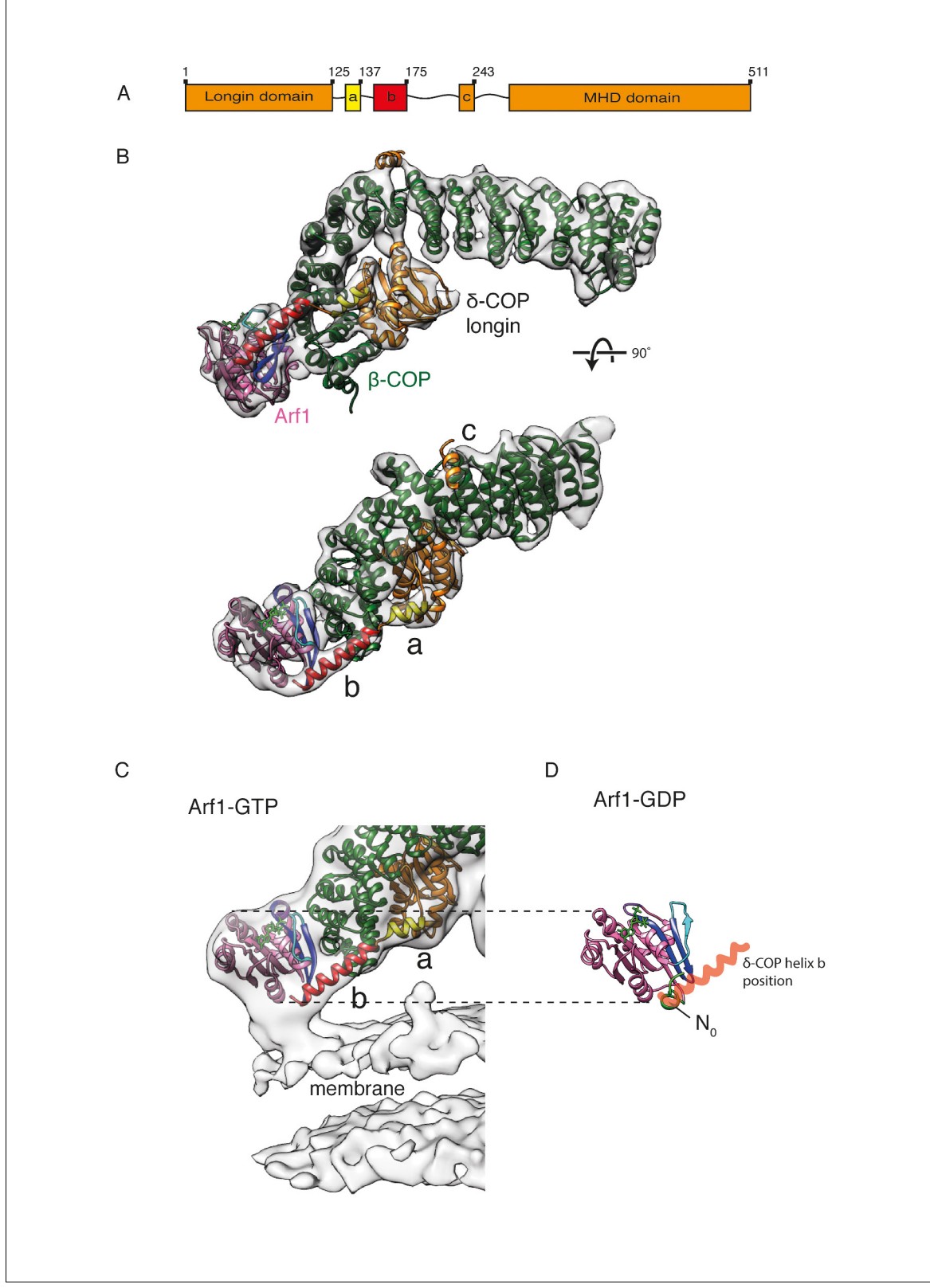

**Figure 5.** δ-COP contains extra α-helices and contacts the Arf1 GTPase. (**A**) A schematic representation of δ-COP domain architecture. Predicted helices downstream of the δ-COP longin domain are labeled 'a', 'b', 'c'. (**B**) Arf1/δ-COP/β-COP complex fitted into the EM density. The MHD δ-COP is not shown here, since it is a part of the inter-triad linkages and is averaged out in the leaf structure. Residues 139–165 of helix b are modeled. Color scheme: EM density – transparent grey, Arf1 - pink, β-COP - dark green, δ-COP - orange. The δ-COP helix 'a' is highlighted in yellow, and helix 'b' in

*Figure 5 continued on next page*

*Figure 5 continued*

red. (**C**) The Arf1/δ-COP/β-COP subcomplex fitted into the EM map illustrated unmasked at a lower isosurface level. (**D**) Arf1 in the GDP-bound state. The orientation of Arf1-GDP is equivalent to that in panel C. The position occupied by the δ-COP helix b in the assembled coat is shown in semi-transparent red: it overlaps with helix N0. Color code: GDP – green, Arf1 – pink, Switch I - cyan, Switch II – purple, Interswitch – blue, helix N0 – green.

observations are consistent with our structural model for the assembled coat in which helix b inter-acts directly with Arf1, and helix c is bound to β-COP to stabilize the complex.

## A role of δ-COP in regulating coat assembly

δ-COP helix a is analogous to the α5 helix in the AP2 μ2-subunit linker, and they are positioned similarly near the β-COP/β2-AP trunk. Helix b is a conserved, COPI-specific feature, at a sequence distance similar to that of the 'bind-back' helix in AP2 μ2 (*Arakel et al., 2016*; *Jackson et al., 2010*). Arakel and colleagues proposed two models to explain a requirement of helix b for retrieval of HDEL/KDEL proteins (*Arakel et al., 2016*). In a first model, this helix was proposed to bind back into a furrow in β-COP, thereby preventing destabilization of the β-COP α-solenoid. In a second model, helix b, which is amphipathic in nature, binds to the membrane, bringing coatomer into close proximity where it can more easily interact with retrieval signals. Our EM structure confirms that helix b is located proximally to the membrane (*Figure 5C*), however the contact with the membrane is small and involves only the C-terminal end of the helix. The distance between the membrane and the nearest conserved hydrophobic residue is too large for a direct contact. This argues against it being a conventional membrane-inserting amphipathic helix (see *Figure 5C*). Our structure suggests that the main interaction partner of helix b is Arf1.

δ-COP helix b interacts directly with the Arf1 Switch I region, Interswitch and C-terminal helix (*Figure 5B*) (in contrast to the interactions observed between AP1 μ1 subunit and Arf1, which do not involve the switch regions of Arf1 [*Shen et al., 2015*]). δ-COP helix b binds the surface of Arf1 in a region that is accessible when Arf1 is in the GTP state and the amphipathic N0 helix is inserted into the membrane, but is occupied by the N0 helix when Arf1 is in the GDP state (PDB: 1MR3, [*Amor et al., 1994*]) (*Figure 5D*). The exposure of this region in Arf1-GTP when N0 moves to insert into the membrane may contribute to the nucleotide-dependent recruitment of coatomer. These interactions also reconcile the apparent nucleotide independence of the interaction between isolated δ-COP and NΔ17Arf1 in solution (*Sun et al., 2007*) with the nucleotide dependence of the interaction between δ-COP and Arf1 during coatomer recruitment to membranes (*Figure 6B*): the truncation of the N0 helix in NΔ17Arf1 exposes the δ-COP binding site regardless of Arf's nucleotide state.

These observations suggest a mechanistic model for the role of δ-COP helix b in COPI function. Upon binding of the Arf1 N0 amphipathic helix to the membrane, the binding site for δ-COP is exposed, and the resulting interaction contributes directly to Arf1-dependent coatomer recruitment to the membrane. The interaction of δ-COP may additionally stabilize Arf1 in its active GTP-bound state by binding its Switch and Interswitch regions. The interaction of δ-COP helix b with the Arf1 GTPase, with the membrane, and possibly with cargo confirms and explains its critical role in regulating COPI function.

We note a further implication of this model: GTP-hydrolysis by Arf1 can directly modulate the conformation of δ-COP. Such modulation may be transmitted to other binding partners such as the KDEL cargo receptor, providing a possible route to link coat hydrolysis to cargo binding/release.

## The linkages between triads

We also determined the structures of the linkages between triads as previously described (*Dodonova et al., 2015*) (*Figure 2—figure supplement 2*). The linkage structures are at lower resolution than that of the leaf, so we fitted our leaf structure as a rigid body (see Materials and methods and *Figure 2—figure supplement 2*) to assess which subunits are close to one another in the linkages. Consistent with our previously published structures (*Dodonova et al., 2015*) the central contacts in linkages I and IV are made by the αε-COP subunits, and those in linkage II are made by the δ-COP MHD. Interestingly neither ε-COP nor the δ-COP MHD are essential for COPI function and the combined deletion of both is not lethal for yeast (*Arakel et al., 2016*; *Kimata et al., 2000*).

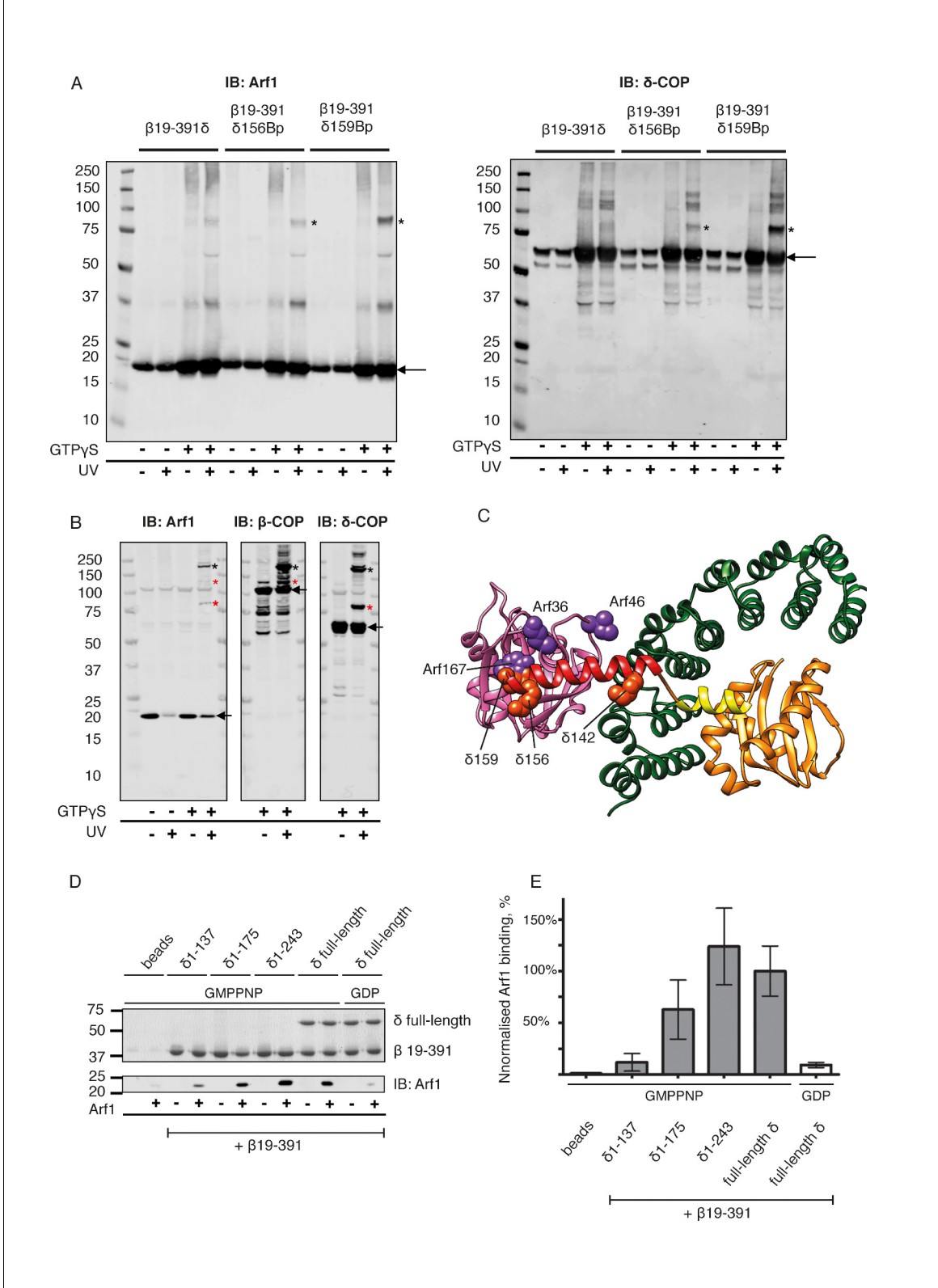

**Figure 6.** Site-directed photo-cross-linking. (**A**) Site-directed photo-cross-linking of chimeric $\beta\delta$ subcomplexes and Arf1-GTP$\gamma$S on Golgi-like liposomes, with photolabile aminoacids (Benzophenone (Bp)) in $\delta$-COP positions 156 or 159 (Ct$\beta$19–391-M$\delta$T156Bp and Ct$\beta$19–391-M$\delta$R159Bp). After irradiation with UV light, cross-linked products were analyzed by SDS-PAGE and western-blot with antibodies directed against Arf1 (left panel) and $\delta$-COP (right panel), and are marked with black asterisks. Bands of non-cross-linked proteins are marked with arrows (IB: immune blot). (**B**) Site-directed photo-cross-

*Figure 6 continued on next page*

Figure 6 continued

linking of Arf1-GTPγS with Bp in both positions 46 and 167 (Arf1-I46BpY167Bp) with coatomer on Golgi membranes. Cross-linked products were analyzed by SDS-PAGE and western blot with antibodies directed against β-COP, δ-COP and Arf1. Black asterisks mark double cross-linked products linked by both photolabile residues, red asterisks mark single cross-linked products linked by Bp at either position 167 or 46 in Arf1. Non-cross-linked proteins are marked with arrows. (C) Ribbon model of the structure of a subcomplex of Ctβ19-391δ1–159 with Arf. Residues involved in cross-linking are shown as spheres (orange-red for δ-COP and purple for Arf1). In summary, photolabile amino acids δ-COP156 and δ-COP159 cross-linked to Arf1, photolabile Arf46 cross-linked to β-COP, and Arf167 cross-linked to δ-COP. Mass-spectrometry cross-linking also identified a cross-link between Arf36 and δ-COP142 (*Dodonova et al., 2015*). (D) Binding of NΔ20CtArf1 GMPPNP to Ctβδ subcomplexes. Subcomplexes contained β19–391-COP and δ-COP including helix a (δ-COP1-137), or δ-COP including helix b (δ-COP1-175), or δ-COP including helix c (δ-COP1-243), or full-length δ-COP, as indicated in the figure. Ctβδ subcomplexes were immobilized on Strep-Tactin sepharose beads. Beads were incubated with purified NΔ20CtArf complexed with GMPPNP or GDP. Pulldowns were analyzed by SDS-PAGE and western-blot. The gels were cut in two pieces. The lower piece was immuno-blotted with antibodies directed against Arf1 (lower panel). The upper part was used for coomassie staining to visualize COP subcomplexes (upper panel) (Note: δ-COP fragments 1–137, 1–175 and 1–243 are not visible in the coomassie stained upper panel as they migrate into the part of the gel that was blotted for quantification of Arf1). (E) Quantification of the data depicted in D. As a control, binding of NΔ20CtArf1 to β19-391COP complexed with full length δ-COP was analyzed in the presence of GDP (last column). Pulldowns were quantified using the Image-Studio software (Li-Cor Bioscience). Quantification was normalized to the βδ-COP subcomplexes containing full-length δ-COP with NΔ20CtArf in its GMPPNP complexed state. (means ± SEM; n = 3). See also *Figure 6—figure supplement 1* and *Figure 6—figure supplement 2*. Source data files for panel E are available in *Figure 6—source data 1*.

The following source data and figure supplements are available for figure 6:

**Source data 1.** Source data file for *Figure 6E*.

**Figure supplement 1.** Mass-spectrometry derived cross-links from newly positioned COPI domains.

**Figure supplement 2.** Site-directed photo-cross-links.

While the contacts formed by these proteins at the linkages may play a regulatory role, they are therefore not essential for COPI function.

The slightly improved resolution of the linkage structures obtained here compared to our previously published maps showed that the density that we interpret as being contributed by a flexible hydrophobic loop of the N-terminal β-propeller of α-COP, is in the vicinity of γ-COP in a neighbouring triad. We found that βArf1 is close to the N-terminal β-propeller of β'-COP in a neighbouring triad, and approaches the trunk domain of β-COP (*Figure 2—figure supplement 2C*). None of these contacts are similar to the previously described crystal contact between the α1 subunit of AP1 and the back side of βArf1 (*Ren et al., 2013*) suggesting that an equivalent interaction is not relevant within the COPI coat. While the low-resolution of the linkage structures precludes more detailed interpretation, the proximity of the peripheral βArf1 molecule to neighbouring triads suggests that βArf1 could modulate inter-triad interactions.

## Interaction of the coat with ArfGAPs

The γArf1 and βArf1 molecules have different interaction partners within the coat (γ-COP, and δ-COP and β-COP, respectively) and are in very different molecular environments (*Figure 2D*). These observations suggest that the two different Arf1 molecules in the coat are differentially regulated. To further investigate the regulation of Arf1 in the coat, we incubated COPI coated vesicles with Arf-GAP1 or ArfGAP2 in the presence of a non-hydrolysable GTP analogue and determined their structures. Previously published biochemical data indicates that almost no ArfGAP1 binds to COPI vesicles generated in vitro in the presence of poorly hydrolysable GTP analogs, while ArfGAP2 is abundant in such vesicle fractions (*Frigerio et al., 2007*). Note, that in the presence of hydrolysable nucleotide GTP, ArfGAP1, 2 and 3 facilitate efficient vesicle uncoating (*Weimer et al., 2008*). We were unable to identify any ArfGAP1 bound to vesicles produced in the presence of GTPγS, while in vesicles incubated with ArfGAP2 we identified an additional density near the central γArf1 molecules in a subset of COPI leaves (*Figure 7A,B* and *Figure 7—figure supplement 1*). No significant additional density was observed near the peripheral βArf1 molecules or at any other position in the structure. The size and shape of the additional mass corresponded very well to the catalytic domain of the ArfGAP2 protein, which we fitted into the density (*Figure 7C*). To further validate this fit, we

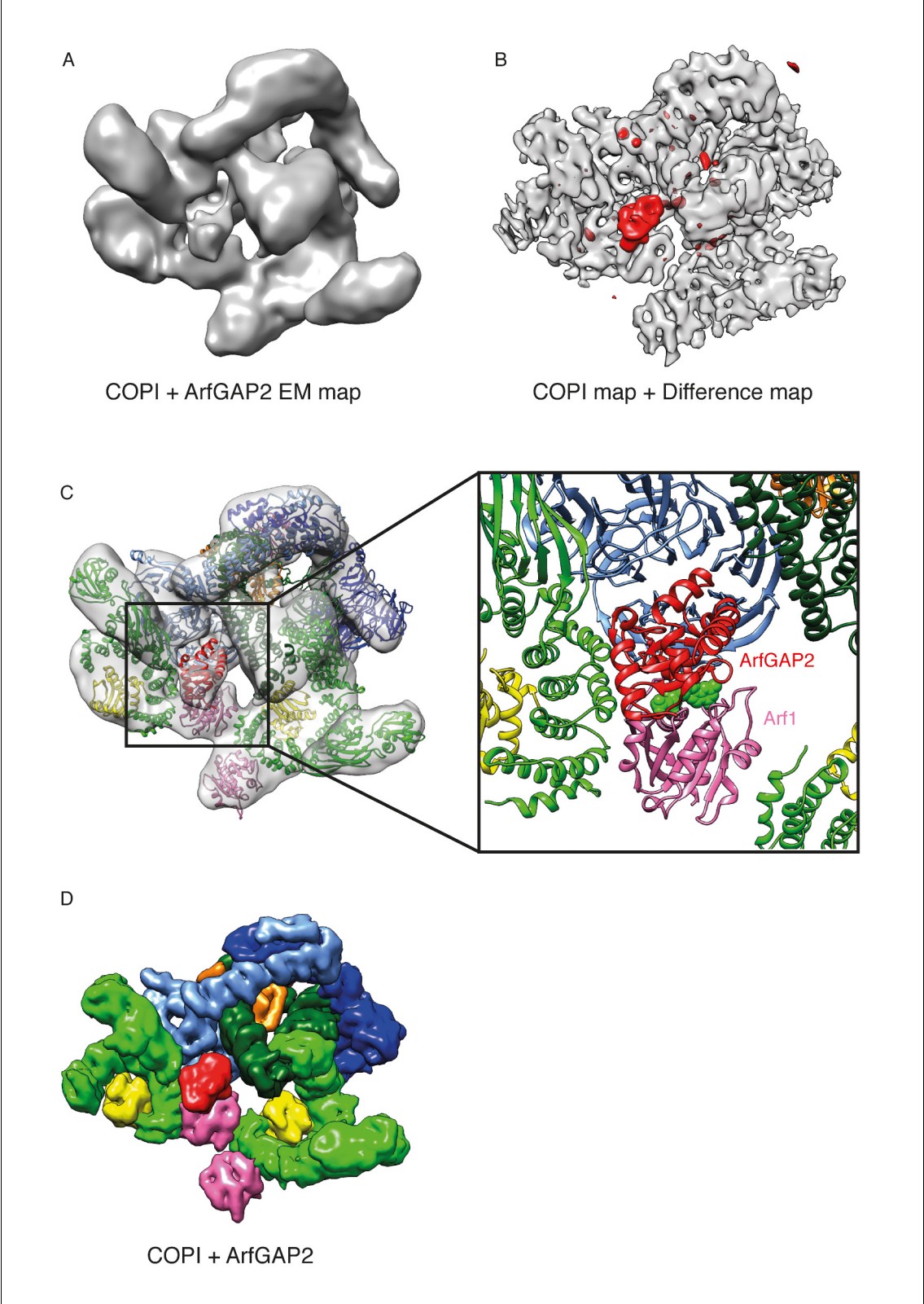

**Figure 7.** Localization of ArfGAP2 within the COPI coat. (**A**) COPI-ArfGAP2 leaf structure as grey isosurface. (**B**) COPI leaf structure (grey) and the difference map between the COPI and the COPI-ArfGAP2 structure (red). (**C**) The fit of COPI and the ArfGAP2 catalytic domain (PDB:2P57) into the COPI-ArfGAP2 leaf structure. The domain is located near the Arf1-nucleotide binding site (nucleotide is shown in green) (**D**) Surface representations of

*Figure 7 continued on next page*

*Figure 7 continued*

the COPI and ArfGAP2 molecular models, illustrating the position of the ArfGAP2 catalytic domain (red) in a niche within the assembled coat. See also *Figure 7—figure supplement 1*.

The following figure supplement is available for figure 7:

**Figure supplement 1.** Position of the ArfGAP2 catalytic domain within in the COPI coat.

superimposed Arf6 from a structure in which it had been co-crystallized with the catalytic domain of the ArfGAP homologue ASAP3 (PDB:3LVQ, [*Ismail et al., 2010*]), with Arf1 in our coat structure. The resulting position of the catalytic domain of ASAP3 coincided with the additional density that we observe (compare *Figure 7* and *Figure 7—figure supplement 1C*). In this structure, the catalytic domain of ArfGAP2 is positioned directly near the Arf1 nucleotide-site, where it can provide the Arginine 'finger' essential for stimulation of GTP-hydrolysis. The observed position of the ArfGAP catalytic domain differs from a previous ArfGAP1-Arf1 structure (*Goldberg, 1999*) in which the Arf-GAP1 catalytic domain is distant from the nucleotide-site of Arf1 (*Figure 7—figure supplement 1D*).

We were not able to resolve the C-terminal, non-catalytic part of ArfGAP2, known to be involved in coatomer binding via interactions with the γ-COP appendage domain (*Kliouchnikov et al., 2009*; *Watson et al., 2004*), most likely because the non-catalytic part of the ArfGAP2 protein is largely disordered and may not form a globular domain that can be positioned at 12 Å resolution (*Pietrosemoli et al., 2013*).

The ArfGAP2 catalytic domain is bound into a niche in the assembled coat formed by Arf1, the γ- and β-COP adaptor subunits, and the β'-COP outer-coat subunit (*Figure 7C and D*). ArfGAP2 activity is increased by 100-fold in the presence of coatomer (*Luo et al., 2009*; *Szafer et al., 2001*). This stimulation requires the presence of both the adaptor subcomplex and outer-coat subcomplex (*Pevzner et al., 2012*), indicating that multiple ArfGAP-coatomer interactions are functionally important. Within a triad, an ArfGAP2-binding niche is formed by β'-COP and β-COP from one leaf and by γ-COP from a neighboring leaf (*Figure 7D*), with the central γArf1 molecules at the bottom. Thus, the complete binding site for the ArfGAP2 catalytic domain is formed when the coat is assembled. This suggests a possible proofreading mechanism: ArfGAP2 recruitment, and the resulting GTP-hydrolysis and coat dissociation can only occur once the coat is assembled, minimizing premature dissociation of coatomer from the membrane.

βArf1 molecules in the triad do not provide an equivalent binding niche for the catalytic domain. We did not observe ArfGAP2 bound near the βArf1 molecules. This is consistent with yeast-two hybrid experiments which showed an interaction between the ArfGAP2 Glo3 and β'-COP and γ-COP but not with other coatomer subunits (*Eugster et al., 2000*). The membrane surface is more exposed near the βArf1 molecules; we suggest that this may facilitate interaction with ArfGAP1 recruited directly to the membrane via its ALPS domains.

## Summary

By combining the βδ-COP crystal structure and the in vitro EM structure of the COPI coat on the vesicle membrane, we have generated a model that reveals molecular details of the coat at the level of protein secondary structure, allowing precise positioning of protein domains and the interpretation of isolated secondary structure elements. The resulting structural model has provided novel functional insight and offers a basis for future concept-driven investigation of molecular mechanisms that underlie vesicular transport.

The two halves of the COPI adaptor subcomplex, γζ-COP and βδ-COP, are thought to have evolved by gene duplication. The structure reveals that they have functionally diverged in two important ways. Firstly, the appendage domains have divergent functions - the γ-appendage sits at the outside of the coat where it provides a binding site for regulatory factors. The β-COP appendage domain links the adaptor and the outer-coat COPI subunits within one coatomer molecule. This functional divergence seems to be mirrored in AP-1 and AP-2, although the details of the interactions are different. Secondly, the two halves of the adaptor subcomplex recruit and position Arf1 in two

different regulatory environments – at the center of the triad, bound to γ-COP (γArf1), and at the periphery of the triad bound to β-COP and δ-COP (βArf1). Our structure shows that the two Arf1 molecules can be differentially regulated both during coatomer recruitment and coat disassembly - βArf1 by interactions of its Switch 1 and N0 helix regions with the essential helix b downstream of the longin domain in δ-COP, and γArf1 by recruitment of ArfGAP2. Differential regulation would allow fine-tuning of coat assembly and cargo recruitment. We speculate that βArf1 molecules may be primarily regulators of coat assembly – modulating coatomer recruitment, dissociation, and the interactions between triads that may be influenced by the presence of cargo, and in which the δ-COP helix b plays a key role. ArfGAP1, bound to the membrane, may function at this stage. We speculate that the γArf1 molecules, via their interaction with ArfGAP2, are the primary regulators of coat disassembly, unlocking the triad and triggering coat collapse upon hydrolysis.

## Materials and methods

### Protein preparation for structural studies

Recombinant *M. musculus* coatomer was expressed and purified from SF9 insect cells (Invitrogen, Karlsruhe). Original SF9 cells were cloned from the parental IPLBSF-21 (Sf-21) cell line that was derived from the pupal ovarian tissue of the fall army worm, *Spodoptera frugiperda*. Invitrogen Sf9 cells were tested by the manufacturer for contamination of bacteria, yeast, mycoplasma and virus and were characterized by isozyme and karyotype analysis. We used a baculoviral expression system essentially as described previously (*Sahlmüller et al., 2011*) with a 'One-Strep-Tag' at the C-terminus of α-COP. Recombinant *S. cerevisiae* myristoylated Arf1, and human nucleotide exchange factor ARNO, were purified from *E. coli* as described previously (*Chardin et al., 1996*; *Randazzo et al., 1992*). Full-length N-terminally His-tagged *R. norvegicus* ArfGAP2 and ArfGAP1 proteins were expressed in insect cells and purified through Ni-NTA chromatography (*Weimer et al., 2008*), and gel filtration.

To determine the most stable domains of coatomer suitable for crystallization, limited proteolysis with subtilisin (Sigma Aldrich, St. Louis Missouri USA) was performed with *C. thermophilum* (Ct) coatomer and subcomplexes. 45–250 μg of the respective complex was treated at various molar ratios. The reaction was incubated for 15 min on ice and then stopped by addition of PMSF to a final concentration of 1 mM. The samples were separated by SDS-PAGE and the resulting fragments were analyzed by MS.

Expression plasmids for truncated forms of Ctβδ-COP subcomplexes were constructed using the pFBDM vector with a One-Strep-Tag fused the N-terminus of β-COP (*Berger et al., 2004*; *Fitzgerald et al., 2006*). Baculoviruses were generated for the subcomplexes Ctβ19-391δ, Ctβ19-391δ1–243, Ctβ19-391δ1–175 and Ctβ19-391δ1–137 by infecting SF9 cells (Invitrogen, Karlsruhe) with recombinant Bacmids prepared using *E.coli* Dh10b MultiBac cells. The dimeric coatomer subcomplexes were produced by co-expression of both subunits in Sf9 insect cells infected with the corresponding baculovirus. Insect cells were harvested 72 hr post infection. Cells were lysed in buffer (25 mM Tris, 300 mM NaCl, 1 mM DTT pH 8.0) using a high pressure Microfluidizer (Microfluidics, Newton USA) and cell debris was pelleted by centrifugation at 100 000 x g for 1 hr. The protein complex was purified using strep-tactin affinity chromatography according to the supplier´s instructions, followed by size exclusion chromatography (Superdex 75 column, GE Healthcare). Purified protein was concentrated using Amicon spin concentrators (Merck Millipore, Darmstadt) and stored at −80°C.

In order to obtain phases by anomalous x-ray diffraction Ctβ19-391δ1–175 subcomplex incorporating selenomethionine (MSE) was produced in Sf9 insect cells. Sf9 cells were grown in *Δ921* Series methionine deficient medium (Expression Systems LLC, USA) supplemented with 150 mg/ml MSE.

### Crystallization, X-ray structure determination, analysis and representation of βδ-COP

Crystallization trials were performed with all construct variants described above and diffracting crystals could be obtained with polyethylene glycols as precipitant with a size range from PEG 3350 to PEG 8000. Best diffracting crystals were obtained with construct Ctβ19-391δ1–175 using sitting drop vapor diffusion in 96-well MRC UVP plates at 18°C with a precipitant and reservoir composition

of 0.2 M magnesium formate, 0.1 M Tris pH 7.0, and 24% PEG3350. Crystallization drops consisted of 400 nl protein solution of Ctβ19-391δ1–175 at a concentration of 13 mg/ml and 400 nl of precipitant. The reservoir volume was 95 μl. Crystals were visible after 10 days. All measured crystals had the orthorhombic space group C222₁ with typical cell dimensions of a = 137.59, b = 177.47, c = 62.72, α=β=γ=90°.

All X-ray diffraction data were collected at ESRF beamlines ID-29 (native) and ID-21 (MSE). Anomalous diffraction data was collected from crystals containing MSE labelled Ctβ19-391δ1–175. All data were integrated using XDS (*Kabsch, 2010*) and scaled with AIMLESS (*Evans, 2006*). Experimental phases were obtained with Phenix. autosol (*Terwilliger et al., 2009*) from anomalous diffraction data and an initial model was obtained using Phenix. autobuild (*Terwilliger et al., 2008*). The final model was built with Coot (*Emsley et al., 2010*) and refined with Phenix.refine (*Afonine et al., 2012*) using the native dataset (see *Table 1*). The asymmetric unit contains one molecule of Ctβ20-390δ1–150. Structure quality was evaluated using MolProbity (*Chen et al., 2010*). Interfaces and crystal contacts were analyzed using PISA (*Krissinel and Henrick, 2007*). Structure figures were created with Chimera (*Pettersen et al., 2004*).

## Pulldown assays

For pulldown assays Arf1 (NΔ20CtArf) was cloned in the petM11 vector, adding an N-terminal HIS₆-TAG, and expressed in *E. coli* BL21 (DE3) cells (Novagen, Merck, Darmstadt). NΔ20CtArf was purified by Ni-NTA affinity chromatography followed by gelfiltration on a Superdex 75 column (GE Healthcare, Buckinghamshire UK) in PD (Pulldown) buffer (25 mM Tris, 150 mM NaCl, 5 mM MgCl₂, 1 mM DTT, pH 7.5).

To test coatomer βδ subcomplexes for binding of Arf1, 200 to 400 μg of the respective subcomplex was immobilized on 50 μl strep-tactin sepharose beads by incubation for one hour at 4°C on a rotary wheel. After immobilization of the subcomplex the beads were washed with PD buffer to remove unbound protein. The soluble form of Arf1-GTP was added to the immobilized subcomplexes and incubated at 4°C for one hour on a rotary wheel. After the incubation the beads were washed extensively with PD buffer. Next the beads were pelleted by centrifugation, resuspended in 50 μl buffer and samples were taken for each subcomplex and mixed with SDS sample buffer. After centrifugation, the proteins in the supernatants were analyzed by SDS-PAGE and western blot.

## Site directed UV-Cross-linking

For photo-cross-linking experiments human Arf1 and yeast N-myristoyl-transferase were cloned in pETDuet vector. Amber stop codons were introduced in Arf1 at positions 46 and 167 by point mutation. Methionine aminopeptidase for improvement of myristoylation efficiency was cloned into pRSF-Duet1 vector. Both plasmids were co-transformed in *E. coli* Bl21 (DE3) cells harbouring the pEVOL plasmid coding for orthogal tRNA recognizing the Amber stop codon and the tRNA synthetase (*Chin et al., 2002*). Cells were grown to OD₆₀₀ of 0.6 at 37°C. After addition of 65 μM of sodium myristate and 1 mM of p-benzoyl-l-phenylalanine, cells were shifted to 27°C. Expression was induced 1 hr after the temperature shift by addition of IPTG (0.5 mM) and arabinose (0.5%) and was continued for 22 hr at 27°C. Photolabile Arf-I46Bp-Y167Bp was purified from the cleared lysate in the presence of 2 mM GDP by size exclusion chromatography on Superdex 200 (GE Healthcare, Buckinghamshire UK) in 25 mM Tris, 150 mM KCl, pH 7.0.

This bivalently photolabile Arf1 was used in a reconstitution assay with liposomes, GTP and complete COPI. For the reaction 5 μg Arf-I46Bp-Y167Bp was mixed with 0.4 μg ARNO, 100 μM GTPγS and 500 μM Golgi-like liposomes in a final volume of 100 μl in 20 mM MOPS pH 7.2, 150 mM KOAc 2 mM Mg(OAc)2, and incubated at 37°C for 10 min. Alternatively 50 μg of Golgi membranes in the presence of 200 mM Sucrose were incubated with 5 μg of the respective photolabile Arf1 with or without 100 μM GTPγS in a final volume of 100 μl buffer at 37°C for 10 min. In a second step, coatomer was added and incubation was resumed at 37°C for 10 min. Membranes were pelleted by centrifugation at 16,000 x g for 30 min at 4°C. Golgi membranes from the liver of male 200 g Wistar rats (*Rattus norvegicus*) were pelleted through 330 μl of 15% (v/v) Sucrose. The pellet was re-suspended in 10 μl buffer and irradiated on ice with 15 × 1 s UV₃₆₆ pulses with 1 s pauses. After irradiation the sample was analysed by MS, SDS-PAGE, and western blot using antibodies directed against β-COP, δ-COP and Arf1. Additionally photolabile amino acids were introduced in position 156 or 159 of δ-

COP (*M. musculus*) using the same approach. These δ-COP variants were introduced in βδ-COP sub-complexes together with Ctβ19–391 producing a chimeric βδ subcomplex. This was done as no anti-bodies were available against Ctδ-COP. The chimeric subcomplexes were tested in the reconstitution assay with liposomes, GTP and Arf1 as described above.

## In vitro budding reaction

Giant Unilamellar Vesicles (GUVs) were prepared by electroformation (*Angelova et al., 1992*) from the Golgi-like lipid mix (*Bigay and Antonny, 2005*). COPI-coated vesicles were produced in vitro by incubating coatomer (840 nM), Arf1 (2 µM), GTPγS (1 mM), ARNO (1.5 µM) and 2 µl GUVs in a total volume of 40 µl for 30 min at 37°C. The budding reaction buffer contained 50 mM HEPES pH 7.4, 50 mM KOAc, 1 mM MgCl$_2$. Protein-A conjugated 10 nm gold was added to the reaction mix in 1:6 vol ratio and the sample was applied onto glow-discharged (30 s, 20 mA) C-flat (Protochips Inc.) multi-hole grids. The grids were blotted from the back side for 11 s at room temperature in a chamber at 85% humidity and plunge-frozen into liquid ethane using a manual plunger.

In order to test activity of the ArfGAP1 and ArfGAP2 proteins, the COPI budding reaction was performed in the presence of GTP, and the reaction mix was incubated for 30 min at 37°C. ArfGAP1, ArfGAP2 or buffer (as a control) were added to the mix in 10 molar excess to coatomer and after 15 min incubation the reaction was plunge-frozen. All samples were imaged in an electron microscope. The control samples contained coated vesicles and buds, whereas the samples incubated with Arf-GAP1 or ArfGAP2 contained only naked liposomes. The functionality of ArfGAP proteins in vitro was also shown previously in budding assays containing rat liver Golgi membranes (*Weimer et al., 2008*).

To explore the structure of the coat in the presence of ArfGAP1 or ArfGAP2, the budding reaction mix was incubated for 30 min prior to addition of a 10 molar excess of ArfGAP1 or ArfGAP2 in the presence of GTPγS nucleotide. The mix was incubated for a further 15–20 min, protein-A conjugated 10 nm gold was added in a 1:6 vol ratio, and the sample was plunge-frozen.

## CryoET sample preparation, data acquisition and initial processing

The plunge-frozen COPI-coated vesicles were imaged in a FEI Titan Krios electron microscope operated at 300 kV and equipped with a Gatan Quantum 967 LS energy filter with a 20 eV energy slit and Gatan K2xp direct electron detector (Gatan Inc.). Tomographic tilt series were acquired with the dose-symmetric tilt-scheme (*Hagen et al., 2017*) over an angular range of ±60° with a 3° increment and a total electron dose of approximately 85 e/Å$^2$. The defocus values ranged from −2.0 to −5.0 um. Data acquisition was controlled using the SerialEM software package (*Mastronarde, 2005*). Five frames were collected in super-resolution and electron-counting mode at each tilt. The super-resolution pixel size at the specimen level was 0.89 Å. The frames were Fourier-cropped, motion-corrected with the K2Align package based on the MotionCorr algorithms (*Li et al., 2013*) and integrated together. Each of the images in the tilt series was low-pass filtered according to the electron-dose acquired by the sample (*Grant and Grigorieff, 2015*). Motion-corrected and dose-filtered tomograms were reconstructed in Imod (*Kremer et al., 1996*).

## Image processing

1733 vesicles and near-complete buds were picked from 61 tomograms. CTF-determination for each individual tilt image was performed using CTFFIND4 (*Rohou and Grigorieff, 2015*). Strip-based CTF-correction and tomogram reconstruction was performed in Imod. Subtomograms were extracted from the surface of the vesicles. Subtomogram averaging was performed using scripts derived from the TOM and Av3 software packages (*Förster et al., 2005*; *Nickell et al., 2005*) and Dynamo (*Castaño-Díez et al., 2012*). Each dataset was split into two halves, each half including odd- and even-numbered vesicles. The COPI triad structure (EMDB-2985, [*Dodonova et al., 2015*]) was low pass filtered to 55 Å and used for the initial alignment step. All further processing steps were performed completely independently on the two half datasets. Preliminary processing was performed using 4x binned data without CTF correction (pixel size 7.12 Å). C3 symmetry was applied. The references were low pass filtered to 35 Å at each alignment step. Iterative rotational and transla-tional alignments were performed until convergence. Next the dataset was cleaned to remove mis-aligned subtomograms based on cross-correlation coefficient threshold, and duplicates were

removed based on the mutual distance between neighboring subtomograms. Final subtomogram alignments were performed on the unbinned (pixel size 1.78 Å) and CTF-corrected data with the low pass filter set to 16 Å. The half datasets contained 19,343 and 19,155 asymmetric units. At the final alignment steps each asymmetric unit of the triad, 'the leaf', representing a COPI molecule and two Arf1 molecules, was processed separately in order to compensate for coat movements and deviation from C3 symmetry. The FSC was calculated for the two final references masked with a soft cylindrical mask and the measured resolution at FSC = 0.143 was 9.4 Å (*Figure 2—figure supplement 1A*).

The local resolution of the EM map was estimated within a small floating window moving within the whole map. The local resolution was highly variable and ranged from 8.7 to 12.7 Å within the protein density (*Figure 2—figure supplement 1B*). We therefore performed multiple local alignments using soft cylindrical masks focused on different parts of the structure. Finally, all the local averages were masked with the corresponding alignment masks, added together, and the map was normalized within the volume enclosed by all local masks. The local resolution ranged from 8.1 to 11.5 Å. The global resolution was 9.2 Å at FSC 0.143. The final structure (*Figure 2B*) was B-factor sharpened (B-factor=−1400) (*Rosenthal and Henderson, 2003*).

## Structural determination of COPI linkages

The COPI triads can be arranged into four distinct types of linkages within the coat (*Faini et al., 2012*). We used the approach described in detail in (*Dodonova et al., 2015*; *Faini et al., 2012*) to determine their positions and orientations. Next we performed subtomogram averaging on each of the linkage datasets to determine their structures (*Figure 2—figure supplement 2*). Each set was processed in two independent halves, which were split based on the odd or even sequential number of the vesicle. C2 symmetry was applied to the linkage III and IV sets. The starting references were obtained by averaging the subvolumes at their initial positions. The alignments were performed iteratively starting with a full search for the in-plane Euler angle and continued with a progressively decreasing angular search step. Duplicate and misaligned subtomograms were removed. Next the subvolumes were extracted at the positions defined during previous iterations from the CTF-corrected unbinned tomograms, and final alignments were performed. The linkage datasets contained 2547, 3312, 140, 1640 asymmetric units from both half sets. The final resolutions of linkages I, II, and IV at FSC 0.143 were respectively 17, 15, 17.3 Å. Linkage III had low abundance in the dataset and the resulting resolution was 31 Å.

In order to generate pseudo-atomic models of the linkages, first, the model of the COPI leaf generated by homology modeling and flexible fitting into the 9.2 Å leaf EM map (see below) was fitted as a rigid body into each of the linkage EM maps. Next, homology models of linkage-specific subunits (α-COP C-terminal domain together with ε-COP and the MHD of δ-COP) were generated in Modeller and fitted as rigid bodies into the central densities in the linkage EM maps.

## Homology modeling and fitting

First the available crystal structures of COPI components were fitted as rigid bodies into the structure of the leaf: Arf1-γζ-COP (PDB:3TJZ [*Yu et al., 2012*]), αβ'-COP (PDB:3MKQ [*Lee and Goldberg, 2010*]), the N-terminal β-propeller of α-COP (PDB:4J87 [*Ma and Goldberg, 2013*]), and the γ-COP appendage (PDB:1PZD [*Hoffman et al., 2003*]). Moreover, several crystal structures corresponding to the densities in the inter-triad linkages were also available: the αε-COP structure (PDB:3MKR and 3MV2 [*Hsia and Hoelz, 2010*; *Lee and Goldberg, 2010*]), the δ-COP MHD (PDB:4O8Q [*Lahav et al., 2015*]).

Homology models of COPI subunits were generated automatically using the HHpred and Modeller servers (*Söding et al., 2005*; *Webb and Sali, 2014*), as described previously (*Dodonova et al., 2015*). The model for β-COP (residues 410–968) was generated based on the structures of the homologous AP2 β2 subunit (PDB:2XA7 and PDB:2G30) (*Edeling et al., 2006*; *Jackson et al., 2010*); the model for γ-COP (residues 312–549) was generated based on the structure of the AP2 α2 (PDB:2XA7) (*Jackson et al., 2010*). α-COP residues 327–813 were modeled based on the β'-COP structure (PDB:3MKQ) (*Lee and Goldberg, 2010*). We performed 'structure-guided' modeling for the C-terminal parts of the β-COP and γ-COP trunk domains, which were absent from the starting x-ray structural models. The γ-COP and β-COP trunk C-terminal parts were modeled as separate

batches of 2–3 helices, which were sequentially fitted into the distinct α-helical EM densities. Loops connecting the batches of helices were added in Modeller.

The structures of all remaining COPI subunits or parts of subunits were available from the PDB and were modeled to represent the *M. musculus* sequence.

Rigid body fitting was performed in Chimera (*Pettersen et al., 2004*). Flexible fitting was performed using NAMD with the MDFF package (*Trabuco et al., 2008*).

### Extra density assignment

After flexible fitting of all COPI subunits into the map, several small regions of the map remained unoccupied. The unoccupied 'extra' densities were identified by subtracting the density occupied by the fitted model from the EM map. The density occupied by the fitted model was generated by the chimera molmap command at 9 Å resolution. The resulting difference map was segmented and the volumes of all separate extra densities were plotted (*Figure 4—figure supplement 1*). The seven large extra densities were numbered according to their sizes (*Figure 4—figure supplement 1*, numbered): '1' near the β'-α-COP interface; '2' near the N-terminal β-propeller domain of α-COP; '3' and '4' near the β-γ-COP trunks interface; '5' near a β-β'-COP contact site; '6' near the β-δ interface; '7' near the Arf1-β-δ interface;

### COPI-ArfGAP2 dataset

The COPI-ArfGAP2 dataset consisted of 27 tomograms containing 690 vesicles and near-complete buds. Subtomogram averaging and alignments were performed for two independent half datasets, which contained either odd- or even-numbered vesicles and consisted of 11,331 and 11,004 asymmetric units respectively. The subtomogram averaging processing pipeline was exactly the same as for the COPI dataset. The final measured resolution of the leaf at the FSC = 0.143 was 9.8 Å. We calculated the difference map between the final COPI-ArfGAP2 EM map and the control COPI EM map. An additional density was visible near the Arf1 molecules located at the center of the triad. Multireference alignment and classification on the complete leaf structure was performed in order to identify the subpopulation of leaves with bound ArfGAP2. To do this, the subtomograms were aligned against the reference from the COPI-ArfGAP2-dataset and from the COPI-dataset and divided into two classes based on cross-correlation. The average structures were generated for both classes. This process was iterated a total of three times. The final COPI-ArfGAP2 class contained 6280 and 6092 asymmetric units in two independent half datasets, comprising in total approximately 65% of the initial dataset. The resolution was 10.1 Å at FSC = 0.143. The second class, which did not contain additional ArfGAP density, consisted of 3497 and 3481 asymmetric units and the resolution at the 0.143 FSC was measured to be 11.7 Å. The final difference map was calculated between the structures produced from the two classes (*Figure 7—figure supplement 1*).

The linkage maps from the COPI-ArfGAP2 were calculated and upon comparison with the control maps did not show any additional densities except that described above.

### COPI-ArfGAP1 dataset

The COPI-ArfGAP1 dataset comprised 24 tomograms containing 833 vesicles and near-complete buds. Subtomogram averaging and alignments were performed for two independent halves of the dataset. The two half datasets contained 14,613 and 14,379 asymmetric units. The subtomogram averaging processing pipeline was exactly the same as for the COPI and COPI-ArfGAP2 datasets. The final measured resolution of the COPI leaf structure at the FSC = 0.143 was 9.6 Å. We calculated the difference map between the final COPI-ArfGAP1 leaf EM map and the control COPI EM map. No large additional densities were observed. The structures of the linkages were calculated and upon comparison with the control maps did not show any additional densities.

## Acknowledgements

EM maps and corresponding structural models are deposited in the Electron Microscopy Data Bank (accession codes EMD-3720, EMD-3721, EMD-3722, EMD-3723, EMD-3724) and the Protein Data Bank (accession codes PDB: 5NZR, 5NZS, 5NZT, 5NZU, 5NZV). Coordinates and structure factors for the X-ray structure of βδ-COP are deposited in PDB under accession code 5MU7. We thank P Schultz for the vectors used for site directed photolabeling, K Bacia for providing *S. cerevisiae* Arf1

protein; A. von Appen and J Kosinski for help with the MS data; F Schur, W Wan, O Avinoam, S Mattei, Y Bykov for discussions. We are grateful to J Goldberg for providing the coordinates of the Arf1-ArfGAP1 model. We thank C Siegmann from the BZH/Cluster of Excellence:CellNetworks crystallization platform for support in protein crystallization. We thank the staff of ESRF and of EMBL-Grenoble for assistance and support in using beamlines ID-21 and ID-29. Our work was technically supported by EMBL IT services, and was funded by the Deutsche Forschungsgemeinschaft within SFB638 (A16) to JAGB and FW, and SFB638 (Z4) to IS; and WI 654/12–1 to FW. FW and IS are investigators of the Cluster of Excellence:CellNetworks.

## Additional information

### Funding

| Funder | Grant reference number | Author |
|---|---|---|
| Deutsche Forschungsgemeinschaft | SFB638 (A16) | Felix Wieland John AG Briggs |
| Deutsche Forschungsgemeinschaft | SFB638 (Z4) | Irmgard Sinning |
| Deutsche Forschungsgemeinschaft | WI 654/12-1 | Felix Wieland |

The funders had no role in study design, data collection and interpretation, or the decision to submit the work for publication.

### Author contributions

SOD, Data curation, Formal analysis, Validation, Investigation, Visualization, Methodology, Writing—original draft, Writing—review and editing; PA, Resources, Formal analysis, Investigation, Visualization, Writing—review and editing; JK, Formal analysis, Investigation, Visualization, Methodology, Writing—review and editing; IG, Resources, Writing—review and editing; SR, Resources, Investigation; WJHH, Investigation, Methodology; IS, Formal analysis, Supervision, Funding acquisition, Methodology, Project administration, Writing—review and editing; FW, Conceptualization, Supervision, Funding acquisition, Project administration, Writing—review and editing; JAGB, Conceptualization, Supervision, Funding acquisition, Investigation, Methodology, Writing—original draft, Project administration, Writing—review and editing

### Author ORCIDs

Svetlana O Dodonova, http://orcid.org/0000-0002-5002-8138
Wim J H Hagen, http://orcid.org/0000-0001-6229-2692
John A G Briggs, http://orcid.org/0000-0003-3990-6910

## Additional files

### Major datasets

The following datasets were generated:

| Author(s) | Year | Dataset title | Dataset URL | Database, license, and accessibility information |
|---|---|---|---|---|
| Dodonova SO, Aderhold P, Kopp J, Ganeva I, Röhling S, Hagen WJH, Sinning I, Wieland F, Briggs JAG | 2017 | The structure of the COPI coat leaf | http://www.ebi.ac.uk/pdbe/entry/emdb/EMD-3720 | Publicly available at the EMBL-EBI Protein Data Bank in Europe (accession no: EMD-3720) |
| Dodonova SO, Aderhold P, Kopp J, Ganeva I, Röhling S, Hagen WJH, Sinning I, Wieland | 2017 | The structure of the COPI coat leaf in complex with the ArfGAP2 uncoating factor | http://www.ebi.ac.uk/pdbe/entry/emdb/EMD-3721 | Publicly available at the EMBL-EBI Protein Data Bank in Europe (accession no: EMD-3721) |

F, Briggs JAG

| | | | | |
|---|---|---|---|---|
| Dodonova SO, Aderhold P, Kopp J, Ganeva I, Röhling S, Hagen WJH, Sinning I, Wieland F, Briggs JAG | 2017 | The structure of the COPI coat linkage I | http://www.ebi.ac.uk/pdbe/entry/emdb/EMD-3722 | Publicly available at the EMBL-EBI Protein Data Bank in Europe (accession no: EMD-3722) |
| Dodonova SO, Aderhold P, Kopp J, Ganeva I, Röhling S, Hagen WJH, Sinning I, Wieland F, Briggs JAG | 2017 | The structure of the COPI coat linkage II | http://www.ebi.ac.uk/pdbe/entry/emdb/EMD-3723 | Publicly available at the EMBL-EBI Protein Data Bank in Europe (accession no: EMD-3723) |
| Dodonova SO, Aderhold P, Kopp J, Ganeva I, Röhling S, Hagen WJH, Sinning I, Wieland F, Briggs JAG | 2017 | The structure of the COPI coat linkage IV | http://www.ebi.ac.uk/pdbe/entry/emdb/EMD-3724 | Publicly available at the EMBL-EBI Protein Data Bank in Europe (accession no: EMD-3724) |
| Dodonova SO, Aderhold P, Kopp J, Ganeva I, Röhling S, Hagen WJH, Sinning I, Wieland F, Briggs JAG | 2017 | The structure of the COPI coat leaf | http://www.ebi.ac.uk/pdbe/entry/pdb/5NZR | Publicly available at the EMBL-EBI Protein Data Bank in Europe (accession no: 5NZR) |
| Dodonova SO, Aderhold P, Kopp J, Ganeva I, Röhling S, Hagen WJH, Sinning I, Wieland F, Briggs JAG | 2017 | The structure of the COPI coat leaf in complex with the ArfGAP2 uncoating factor | http://www.ebi.ac.uk/pdbe/entry/pdb/5NZS | Publicly available at the EMBL-EBI Protein Data Bank in Europe (accession no: 5NZS) |
| Dodonova SO, Aderhold P, Kopp J, Ganeva I, Röhling S, Hagen WJH, Sinning I, Wieland F, Briggs JAG | 2017 | The structure of the COPI coat linkage I | http://www.ebi.ac.uk/pdbe/entry/pdb/5NZT | Publicly available at the EMBL-EBI Protein Data Bank in Europe (accession no: 5NZT) |
| Dodonova SO, Aderhold P, Kopp J, Ganeva I, Röhling S, Hagen WJH, Sinning I, Wieland F, Briggs JAG | 2017 | The structure of the COPI coat linkage II | http://www.ebi.ac.uk/pdbe/entry/pdb/5NZU | Publicly available at the EMBL-EBI Protein Data Bank in Europe (accession no: 5NZU) |
| Dodonova SO, Aderhold P, Kopp J, Ganeva I, Röhling S, Hagen WJH, Sinning I, Wieland F, Briggs JAG | 2017 | The structure of the COPI coat linkage IV | http://www.ebi.ac.uk/pdbe/entry/pdb/5NZV | Publicly available at the EMBL-EBI Protein Data Bank in Europe (accession no: 5NZV) |

The following previously published datasets were used:

| Author(s) | Year | Dataset title | Dataset URL | Database, license, and accessibility information |
|---|---|---|---|---|
| Dodonova SO, Diestelkoetter-Bachert P, von Appen A, Hagen WJ, Beck R, Beck M, Wieland F, Briggs JA | 2015 | The structure of the COPI coat triad | http://www.ebi.ac.uk/pdbe/entry/emdb/EMD-2985 | Publicly available at the EMBL-EBI Protein Data Bank in Europe (accession no: EMD-2985) |
| Ma W, Goldberg J | 2013 | Crystal structure of alpha-COP | http://www.ebi.ac.uk/pdbe/entry/pdb/4J87 | Publicly available at the EMBL-EBI Protein Data Bank in Europe (accession no: 4J87) |
| Amor JC, Harrison DH, Kahn RA, | 1994 | Saccharomyces cerevisiae ADP-ribosylation Factor 2 (ScArf2) | http://www.ebi.ac.uk/pdbe/entry/pdb/1MR3 | Publicly available at the EMBL-EBI Protein |

| | | | | |
|---|---|---|---|---|
| Ringe D | | | complexed with GDP-3'P at 1.6A resolution | Data Bank in Europe (accession no: 1MR3) |
| Ismail SA, Vetter IR, Sot B, Wittinghofer A | 2010 | | The crystal structure of ASAP3 in complex with Arf6 in transition state | http://www.ebi.ac.uk/pdbe/entry/pdb/3LVQ | Publicly available at the EMBL-EBI Protein Data Bank in Europe (accession no: 3LVQ) |
| Yu X, Breitman M, Goldberg J | 2012 | | Crystal Structure of Arf1 Bound to the gamma/zeta-COP Core Complex | http://www.ebi.ac.uk/pdbe/entry/pdb/3TJZ | Publicly available at the EMBL-EBI Protein Data Bank in Europe (accession no: 3TJZ) |
| Lee C, Goldberg J | 2010 | | Crystal structure of yeast alpha/betaprime-COP subcomplex of the COPI vesicular coat | http://www.ebi.ac.uk/pdbe/entry/pdb/3MKQ | Publicly available at the EMBL-EBI Protein Data Bank in Europe (accession no: 3MKQ) |
| Hoffman GR, Rahl PB, Collins RN, Cerione RA | 2003 | | Structural Identification of a conserved appendage domain in the carboxyl-terminus of the COPI gamma-subunit | http://www.ebi.ac.uk/pdbe/entry/pdb/1PZD | Publicly available at the EMBL-EBI Protein Data Bank in Europe (accession no: 1PZD) |
| Lee C, Goldberg J | 2010 | | Crystal structure of yeast alpha/epsilon-COP subcomplex of the COPI vesicular coat | http://www.ebi.ac.uk/pdbe/entry/pdb/3MKR | Publicly available at the EMBL-EBI Protein Data Bank in Europe (accession no: 3MKR) |
| Hsia K, Hoelz A | 2010 | | Crystal Structure of a-COP in Complex with e-COP | http://www.ebi.ac.uk/pdbe/entry/pdb/3MV2 | Publicly available at the EMBL-EBI Protein Data Bank in Europe (accession no: 3MV2) |
| Lahav A, Rozenberg H, Parnis A, Cassel D, Adir N | 2015 | | Crystal structure of bovine MHD domain of the COPI delta subunit at 2.15 A resolution | http://www.ebi.ac.uk/pdbe/entry/pdb/4O8Q | Publicly available at the EMBL-EBI Protein Data Bank in Europe (accession no: 4O8Q) |
| Jackson LP, Kelly BT, McCoy AJ, Gaffry T, James LC, Collins BM, Honing S, Evans PR, Owen DJ | 2010 | | AP2 CLATHRIN ADAPTOR CORE IN ACTIVE COMPLEX WITH CARGO PEPTIDES | http://www.ebi.ac.uk/pdbe/entry/pdb/2XA7 | Publicly available at the EMBL-EBI Protein Data Bank in Europe (accession no: 2XA7) |
| Edeling MA, Mishra SK, Keyel PA, Steinhauser AL, Collins BM, Roth R, Heuser JE, Owen DJ, Traub LM | 2006 | | Beta appendage of AP2 complexed with ARH peptide | http://www.ebi.ac.uk/pdbe/entry/pdb/2G30 | Publicly available at the EMBL-EBI Protein Data Bank in Europe (accession no: 2G30) |
| Tong Y, Dimov S, Shen L, Zhu H, Tempel W, Landry R, Arrowsmith CH, Edwards AM, Sundstrom M, Weigelt J, Bochkarev A, Park H | 2007 | | GAP domain of ZNF289, an ID1-regulated zinc finger protein | http://www.ebi.ac.uk/pdbe/entry/pdb/2P57 | Publicly available at the EMBL-EBI Protein Data Bank in Europe (accession no: 2P57) |

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
