## [Decision Letter]

Thank you for submitting your article "9Å structure of the COPI coat reveals diverged functions of γζ-COP-Arf1 and βδ- COP-Arf1" for consideration by *eLife*. Your article has been reviewed by three peer reviewers, and the evaluation has been overseen by a Reviewing Editor and Randy Schekman as the Senior Editor. The following individual involved in review of your submission has agreed to reveal his identity: Benjamin D Engel (Reviewer #3).

The reviewers have discussed the reviews with one another and the Reviewing Editor has drafted this decision to help you prepare a revised submission.

Summary:

The authors have used state-of-the-art cryo-EM tomography, combined with X-ray crystallography and photo-crosslinking, to determine the structure of the COPI coat at 9Å resolution. The structure provides many mechanistic insights, not only into the workings of the COPI coat, but also into how the related AP complex-containing coats might function, and into how the whole family may have evolved. It will be of interest to both structural and cell biologists – including evolutionary cell biologists.

Essential revisions:

All three reviewers acknowledge the excellence and importance of the work. Although a few additional experiments are suggested, in every case the authors could address the reviewers' comments by making modifications to the text (e.g. by softening interpretations), unless they have the data already in hand. Many of the reviewers' suggestions are about ways to make an inherently complicated structure as clear as possible to the reader. These are constructive comments that the authors should take on board, so that the paper can be understood by as many readers as possible. Other comments include concerns about a few discrepancies, and about the speculative nature of their model, which the authors should easily be able to address in the text. The reviewers have done such a thorough job that instead of trying to paraphrase their comments, I am appending their complete reviews below.

Reviewer #1:

Vesicle coat complexes enable vesicular traffic between different compartments of the endomembrane system. They are highly dynamic macromolecular complexes that reversibly associate with membranes, select protein and lipid cargo, orchestrate vesicle budding and scission from the donor compartment, and – via the interaction with tethering complexes – contribute to the selection of the acceptor compartment. All these different steps must be properly coordinated and proofread to ensure smooth operation of vesicular traffic, which is a fundamental cellular process. Structural insight into some vesicle coat complexes such as clathrin or COPII is very far advanced whereas the structural analysis of COPI and its conformational dynamics has lagged behind for many years.

The manuscript by Dodonova et al., is an impressive illustration of how the combination of cutting edge cryo-electron tomography methods with X-ray crystallography and elegant functional experiments can bring about a major leap in understanding. This work intrigues and fascinates by providing many deep insights into the mechanics of the COPI coat.

Specifically, the following major advances are presented:

The manuscript finally closes a gap by providing a higher-resolution structure of β-δ-COP. As evident from the previous ET paper by Dodonova et al., such crystal structures are important to validate ET structures and to harness their full potential. Homology models of COPI subunits based on clathrin adaptor structures were of limited use in this context and the data presented in this paper elevates the resulting structural model of the entire COPI complex on the membrane to a stunning level of detail and quality.

Next, the manuscript makes substantial progress in understanding the structural integration of the appendage domains of β- and γ-COP respectively. The insights provided into their different accessibility and packing with other subunits in the case of the β-COP appendage are truly striking and will be key to a full functional understanding of the COPI vesicle formation.

Finally, the manuscript is of fundamental importance because it provides detailed insights into COPI's functional interplay with the small GTPase Arf1. This interaction is crucial for COPI recruitment to the membrane, coat formation, and uncoating. The data reveal two structurally and functionally distinct modes by which the small GTPase contacts COPI. This functional angle of the manuscript is developed a lot further by presenting yet another structure including the catalytic domain ArfGAP2.

Given the array of novel and exciting findings one aspect is treated with surprising lack of depth and rigorousness: The issue of the linkages in the COPI coat. The authors start the short description of the pertinent data and their interpretation by saying "We determined the structure of the linkages between triads as previously described (Dodonova, 2015)", suggesting that the following sentences logically build on the previous paper. But they don't. In the current model other subunits are delineated as being important for the linkages between triads. Also, work demonstrating that yeast remains viable without epsilon-COP and the mu-homology domain (Arakel et al., 2016) is not discussed in this context. It is clearly communicated that the resolution of the structure is lower in the relevant regions but a comparison between the previous paper by Dodonova et al., and their current data is warranted. A more conceptual discussion of why the COPI coat may be so different compared to COPII and clathrin in terms of the regularity of its structure would be very useful to the field.

A control experiment for Panel 6A would be important. We agree with the assignment of the specific cross-linked products in the anti-δ-COP IB. However, in the anti-Arf1 IB on the left, the difference between lanes 4 and 8 is very subtle, indeed. It would be an important corroboration of this experiment if a mutant form of δ-COP where the hydrophobic side chains of the helix are mutated were tested as a negative control.

Reviewer #2:

This manuscript by Dodonova et al., presents the highest resolution structure to date of the COPI coat by fitting a new 2.6 Å crystal structure of β-δ-COP, existing atomic structures of other COP proteins, and homology models of the remaining components into a new 9 Å subtomogram average of the membrane-bound COPI coat reconstituted on membranes in vitro. The authors then build upon this structure with a subtomogram average of COPI bound to ArfGAP2, which binds only one of the two Arf1 sites, providing insights into the mechanisms of COPI coat assembly and disassembly.

While these findings are somewhat incremental, they are important and greatly increase our structural understanding of COPI function. Moreover, the work is technically excellent; a sub-nanometer subtomogram average of a relatively flexible and modular complex is certainly at the leading edge of what is possible with current technology. The jump in resolution from 13 Å (Dodonova et al., Science 2015) to 9 Å (this study) is biologically significant because it allows the secondary structure of the assembled COPI triad to be resolved and fairly accurately modeled for the first time.

I have no critical concerns that would preclude the publication of this study. The structures presented here are of high quality and will be very useful to the scientific community. However, the issues below must be addressed:

1) I do not believe the data in this paper supports the model proposed by the authors for the recruitment of coatomer by Arf1: "These observations suggest a mechanistic model for the role of δ-COP helix b in COPI function. Upon binding of the Arf1 N0 amphipathic helix to the membrane, the binding site for δ-COP is exposed, and the resulting interaction contributes directly to Arf1-dependent coatomer recruitment to the membrane."

The authors state in the introduction that Arf1-GDP is recruited to the membrane, where it is activated into Arf1-GTP by a GEF protein. Then Arf1-GTP recruits the coatomer via an interaction with β-δ-COP. The low binding affinity of Arf1-GDP for β-δ-COP is confirmed by the pulldown assays in Figure 6 to E, conflicting with the author's model that Arf1-GDP recruits the coatomer. I agree that the amphiphatic NO helix of Arf1-GDP (which is absent from Arf1-GTP) likely helps recruit Arf1-GDP to the membrane. However, the less conflicting model to propose would be that a GEF protein then converts it into Arf1-GTP, which removes the NO helix, enabling the "b" helix of δ-COP to bind in the site that was previously occupied by the NO helix. Interestingly, while losing the amphiphatic N0 helix could weaken the association of Arf1-GTP with the membrane, the "b" helix of δ-COP is also amphiphatic and so may be able to compensate and stabilize the membrane interaction (although the authors state that the nearest hydrophobic residues of the "b" helix are too far from the membrane, the "b" helix does occupy the same location as the NO helix).

If the authors want to propose a model where membrane binding by Arf1-GDP increases the affinity of Arf1-GDP for δ-COP, thus recruiting coatomer to the membrane prior to GEF-mediated conversion to Arf1-GTP, then they must do this experiment or provide strong supporting references.

2a) The mouse sequences for β-COP and δ-COP (mouse proteins were used for the EM structure) are 953 and 511 amino acids long, respectively, according to NCBI and Figure 5. However, the structure presented in Figure 1 as the "crystal structure of the β-δ-COP subcomplex" only contains β-COP 19-391 and δ-COP 1-175 from C. thermophilium (the COP proteins are highly conserved, so mixing organisms in the model should be fine). This is only 39% of β-COP and 34% of δ-COP. While this is not a problem in itself, the title of the Results section and the title of the figure legend should state that this is a "partial structure of β-δ-COP" or a "structure of the interaction between β-COP and δ-COP". In order to clearly convey which regions of the proteins are included in the structure, an inset should be added to Figure 1 that diagrams the domains of β-COP and δ-COP (similar to Figure 5), illustrating which parts are in the crystal structure and which parts are not.

2b) Figure 5 returns to the structure of β-δ-COP, this time looking at the subtomogram average to show the interaction with Arf1. Compared to the crystal structure in Figure 1, more (all?) of β-COP is displayed. However, the C-terminal half of δ-COP, corresponding to the MHD domain, is still absent. Later in the manuscript, it is briefly mentioned that the δ-COP MHD domain is part of the linkage between COPI triads (along with epsilon-COP and the C-terminus of α-COP; Figure 6—figure supplement 2, and crucially, only in the figure legend of Figure 2—figure supplement 2 is it mentioned that "δ-COP MHD is located in the linkage, and not in the COPI leaf, so it is not enclosed by the EM density". Finally, I understood why the MHD domain is not pictured in Figure 5, but this took far too much detective work. Thus: 1) the legend for Figure 2 should state which COP domains are not resolved within the 9A leaf structure due to their location in the linkage region outside of the triad, and 2) the legend of Figure 5 should reiterate that the MHC domain is not shown because it is outside the leaf structure.

3) Related to the above issue, additional labels should be added to many of the figures in order to make it easier to understand which components are being displayed in the images without having to continually return to the Figure 2 legend for the color code. If only specific domains are being shown instead of full proteins, this should also be labeled. This will make the results much easier to follow for readers who are not intimately familiar with the COPI structure. Specifically:

a) Color-coded labels on figure panels adjacent to the corresponding subunits:

Figure 1: dark green "β-COP" (in addition to existing labels); Figure 3: red "appendage sandwich subdomain", purple: "appendage platform subdomain"; Figure 3: black "γ-COP appendage", light blue "β'-COP"; Figure 3: black "β-COP appendage", dark blue "α-COP"; Figure 4: light blue "β'-COP", dark blue "α-COP"; Figure 4: dark blue "α-COP N-terminus"; Figure 4: dark green "β-COP", orange "δ-COP", light green "γ-COP", yellow "zeta-COP", pink "Arf1"; Figure 5: dark green "β-COP", black "Longin domain" (in addition to "a", "b", "c"); Figure 7: red "ArfGAP2", pink "Arf1".

b) The degree of rotation (e.g. 90°) should be added next to the rotation symbols in each of the Figure 2, Figure 3, Figure 4, Figure 5, Figure 2—figure supplement 2 and 2D Figure 7—figure supplement 1. Additional rotation symbols and degrees should be added to Figure 4.

c) In Figure 5, it would be very helpful to illustrate the position of the δ-COP "b" helix from the cryo-EM structure relative to Arf1-GDP in order to show how the N0 amphiphatic helix of Arf1-GDP blocks the binding site for δ-COP on Arf1. Since there is space, this could either be incorporated into the current image or be shown in a second image.

d) Figure 6 would be much more clear if the "a" and "b" helices of δ-COP were colored red and yellow, respectively, to be consistent with their coloration in Figure 1 and Figure 5, and Figure 1—figure supplement 1 Upon doing so, the yellow spheres representing crosslinked residues should be recolored to match the red "b" helix (perhaps a dark orange would look good).

4) No PDB structures are listed for the atomic models that were fit into the EM density (α-COP, Arf1-GTP, etc.). While these structures may have been named in the previous Science paper, at least in the methods of the current manuscript, these structures should be listed and the appropriate papers should be cited.

5) As in situ structures within cells are now possible, it should be made more explicitly clear in the abstract, introduction, and summary that the structure being presented is from an in vitro reconstituted system. Specifically:

Abstract: change "we determined the structure of the COPI coat assembled on membranes" to something like "we determined the structure of the COPI coat assembled on membranes in vitro".

Introduction: change "we recently determined the architecture of the COPI coat assembled on the vesicle membrane" to something like "we recently determined the architecture of the in vitro reconstituted COPI coat assembled on vesicle membranes".

Summary: change "by combining the β-δ-COP crystal structure and the EM structure of the COPI coat on the vesicle membrane" to something like "by combining the β-δ-COP crystal structure and the in vitro EM structure of the COPI coat on the vesicle membrane".

Reviewer #3:

General assessment

Dodonova and colleagues have determined a new crystal structure of part of the β/δ-COP subcomplex, which revealed several interesting new features in the 'linker region' of δ-COP (between the longin domain and μ-homology domains). They use this crystal structure, together with published X-ray structures, homology models, and their cryo-ET data, to generate an improved model of the mammalian COPI coat at ~9Å resolution using vesicles formed in vitro. The improved resolution allows them to locate and place key COPI subunits within coats assembled on membranes, including the β and γ appendage domains, and important helices in δ- COP. Photo cross-linking data support their structural model that places Arf1 in close proximity to β/δ-COP and specifically near to the δ-COP helix b. The authors also present structural evidence that Arf1 molecules in the coat are not equivalent and that only specific Arfs (bound to γ-COP) would be accessible for ArfGAP2 binding.

Overall, the paper and associated movie presents beautiful new crystallographic and EM data that provide insight into the overall structure of the COPI coat. But the paper currently has significant weaknesses. It is not written especially well: it would 'read' much better if the authors clearly separated results from discussion and speculation, and several important discussion aspects seem to be missing (e.g. a more complete comparison of COPI versus AP2 appendage domains) or glossed over. The figures could be significantly improved in terms of color selection, consistency, and labelling. If you're not already very familiar with these structures, you will likely struggle to understand this story. There is a great deal of speculation here.

There are two weaknesses in this story. Many questions remain around the biochemical data, which is currently insufficient to demonstrate how δ-COP helices interact with Arf1. This seems to be a key interaction in the coat that unites membrane recruitment, cargo recognition, and coat assembly. The pulldown data also do not seem to support placement of δ-COP helixC in the electron density.

Secondly, they emphasize the possibility and importance of non-equivalent Arfs in the coat but provide no biochemical or in vitro data to support this. The structure may imply two different functions, but they have certainly not shown it. One would prefer to see specific structure-based point mutants in the β/δ and/or γ/ζ sub-complexes (or perhaps just in the δ-COP linker?) used in liposome budding assays. One might predict mutations in the δ-COP linker alone or in the β/δ-COP interface would result in inefficient coat recruitment or budding, while mutations in γ/ζ subcomplexes would not. Finally, coat disassembly is only one model for ArfGAP function; yeast data suggest a role in coat assembly too. The authors should comment on based on their new structure.

Major comments:

1. δ-COP helices section. Does it make sense for helixC to be assigned to this position in the density? It seems located far away: what's the distance the δ-COP unstructured linker must span? And where does that imply the MHD sits? Why is helix c needed to help coordinate the MHD? This is very speculative, and the biochemical data don't seem to support the placement of helixC (below).

2) The authors need to describe more clearly experimental details for pulldowns between β/δ-COP and Arf1 (GMPPNP) in the main text and figure labels (ie. strep-tagged sub-complexes on sepharose). Importantly, why don't we see two Coomassie bands in all lanes of Figure 6, corresponding to β- and δ-COP? And why haven't the authors shown pulldowns with both GDP- and GTP-locked forms of Arf1 if they say the interaction is nucleotide 'agnostic'?

Although the pulldowns demonstrate a biochemical interaction between purified β-/δ-COP and Arf1, such an interaction has been shown previously by Jonathan Goldberg. What's new here seems to be the importance of the linker region for the interaction. The authors state the main interaction partner of δ-COP helix b is Arf1, but their pulldown data indicate helix C is important (Figure 6). But then they place helixC very far away from Arf1 in the density (Figure 6). Why not do pulldowns with δ-COP linkers (of different lengths) and Arf1 in both nucleotide states to nail down this interaction? Is the affinity too low, even by Western blotting? Either way, the biochemistry suggests helixC is located much closer to Arf1. This story is interesting but is 'left hanging'.

3) The authors claim δ-COP helix b makes interactions with specific Arf1 regions (Switch I/Interswitch). I'm not convinced the resolution is sufficient to make these claims. And they have not shown us biochemical data to support the idea that δ-COP is agnostic to the nucleotide state of Arf1.

4) The authors propose that only the Arfs near γ-COP would be hydrolyzed by ArfGAP2? They suggest only these γArfs would be accessible for ArfGAP2 binding while βArfs would facilitate interaction with ArfGAP1. Is there evidence somewhere that supports the idea that different ArfGAPS bind different coatomer subunits (yeast two hybrid etc)? And is there any relation at all between one of these Arfs and the so-called 'backside' Arf1 seen in Hurley's open AP1 structure?

5) The idea of non-equivalent Arfs could be tested biochemically by combining structure-based mutagenesis and their liposome budding assay to count budding vesicles per area. This would be an important confirmation of their structure-based hypothesis.

---

## [Author Response]

Reviewer #1:

*Vesicle coat complexes enable vesicular traffic between different compartments of the endomembrane system. They are highly dynamic macromolecular complexes that reversibly associate with membranes, select protein and lipid cargo, orchestrate vesicle budding and scission from the donor compartment, and – via the interaction with tethering complexes – contribute to the selection of the acceptor compartment. All these different steps must be properly coordinated and proofread to ensure smooth operation of vesicular traffic, which is a fundamental cellular process. Structural insight into some vesicle coat complexes such as clathrin or COPII is very far advanced whereas the structural analysis of COPI and its conformational dynamics has lagged behind for many years.*

*The manuscript by Dodonova et al., is an impressive illustration of how the combination of cutting edge cryo-electron tomography methods with X-ray crystallography and elegant functional experiments can bring about a major leap in understanding. This work intrigues and fascinates by providing many deep insights into the mechanics of the COPI coat.*

*Specifically, the following major advances are presented:*

*The manuscript finally closes a gap by providing a higher-resolution structure of β-δ-COP. As evident from the previous ET paper by Dodonova et al., such crystal structures are important to validate ET structures and to harness their full potential. Homology models of COPI subunits based on clathrin adaptor structures were of limited use in this context and the data presented in this paper elevates the resulting structural model of the entire COPI complex on the membrane to a stunning level of detail and quality.*

*Next, the manuscript makes substantial progress in understanding the structural integration of the appendage domains of β- and γ-COP respectively. The insights provided into their different accessibility and packing with other subunits in the case of the β-COP appendage are truly striking and will be key to a full functional understanding of the COPI vesicle formation.*

*Finally, the manuscript is of fundamental importance because it provides detailed insights into COPI's functional interplay with the small GTPase Arf1. This interaction is crucial for COPI recruitment to the membrane, coat formation, and uncoating. The data reveal two structurally and functionally distinct modes by which the small GTPase contacts COPI. This functional angle of the manuscript is developed a lot further by presenting yet another structure including the catalytic domain ArfGAP2.*

*Given the array of novel and exciting findings one aspect is treated with surprising lack of depth and rigorousness: The issue of the linkages in the COPI coat. The authors start the short description of the pertinent data and their interpretation by saying "We determined the structure of the linkages between triads as previously described (Dodonova, 2015)", suggesting that the following sentences logically build on the previous paper. But they don't. In the current model other subunits are delineated as being important for the linkages between triads. Also, work demonstrating that yeast remains viable without epsilon-COP and the mu-homology domain (Arakel et al., 2016) is not discussed in this context. It is clearly communicated that the resolution of the structure is lower in the relevant regions but a comparison between the previous paper by Dodonova et al., and their current data is warranted. A more conceptual discussion of why the COPI coat may be so different compared to COPII and clathrin in terms of the regularity of its structure would be very useful to the field.*

We have expanded our discussion of the linkages and added further panels to Figure 2—figure supplement 2. We now clarify that we observe the same subunits contributing to the linkages as previously described, but also some additional ones. We have discussed the observation that epsilon and the δ MHD are not essential.

We have not added a conceptual discussion of why COPI may be different to COPII and clathrin. In particular, in the clathrin field, there is currently limited information available on the structural relationship between adaptor and coat subunits, and we hope that ongoing work in this field will soon permit a more informed comparison.

*A control experiment for Panel 6A would be important. We agree with the assignment of the specific cross-linked products in the anti-δ-COP IB. However, in the anti-Arf1 IB on the left, the difference between lanes 4 and 8 is very subtle, indeed. It would be an important corroboration of this experiment if a mutant form of δ-COP where the hydrophobic side chains of the helix are mutated were tested as a negative control.*

We agree with reviewer that in Figure 6 the difference between δ-COP156Bp and control (lane 4) is subtle. The faint band in Figure 6 lane 4 may be due to a UV-crosslink between aromatic amino acids within Arf and δ-COP, independent of Benzophenone. The difference in intensity between δ-COP159Bp and the control is much stronger however, consistent with the orientations of side chain 159 and side chain 156 as depicted in Figure 6, where position 159 is more closely oriented towards Arf1. The interaction is further validated by the photo-crosslinking assays using photolabile amino acids in Arf1, by the mass-spec cross-linking data, and by the experiments assessing Arf1 binding to δ mutants lacking the helix.

*Reviewer #2:*

*1) I do not believe the data in this paper supports the model proposed by the authors for the recruitment of coatomer by Arf1: "These observations suggest a mechanistic model for the role of δ-COP helix b in COPI function. Upon binding of the Arf1 N0 amphipathic helix to the membrane, the binding site for δ-COP is exposed, and the resulting interaction contributes directly to Arf1-dependent coatomer recruitment to the membrane."*

*The authors state in the introduction that Arf1-GDP is recruited to the membrane, where it is activated into Arf1-GTP by a GEF protein. Then Arf1-GTP recruits the coatomer via an interaction with β-δ-COP. The low binding affinity of Arf1-GDP for β-δ-COP is confirmed by the pulldown assays in Figure 6 to E, conflicting with the author's model that Arf1-GDP recruits the coatomer. I agree that the amphiphatic NO helix of Arf1-GDP (which is absent from Arf1-GTP) likely helps recruit Arf1-GDP to the membrane. However, the less conflicting model to propose would be that a GEF protein then converts it into Arf1-GTP, which removes the NO helix, enabling the "b" helix of δ-COP to bind in the site that was previously occupied by the NO helix. Interestingly, while losing the amphiphatic N0 helix could weaken the association of Arf1-GTP with the membrane, the "b" helix of δ-COP is also amphiphatic and so may be able to compensate and stabilize the membrane interaction (although the authors state that the nearest hydrophobic residues of the "b" helix are too far from the membrane, the "b" helix does occupy the same location as the NO helix).*

*If the authors want to propose a model where membrane binding by Arf1-GDP increases the affinity of Arf1-GDP for δ-COP, thus recruiting coatomer to the membrane prior to GEF-mediated conversion to Arf1-GTP, then they must do this experiment or provide strong supporting references.*

We apologize that we were not sufficiently clear. Indeed, exactly as the reviewer suggests, our model is that in Arf1-GTP, the N0 helix is moved from its location allowing δ-COP to bind in the site previously occupied by N0. In the GTP state the N0 helix is then inserted into the membrane. We have edited the text to clarify this:

“δ-COP helix b binds the surface of Arf1 in a region that is accessible when Arf1 is in the GTP state and the amphipathic N0 helix is inserted into the membrane, but is occupied by the N0 helix when Arf1 is in the GDP state (PDB: 1MR3, (Amor et al., 1994)) (Figure 5). The exposure of this region in Arf1-GTP when N0 moves to insert into the membrane may contribute to the nucleotide-dependent recruitment of coatomer.”

*2a) The mouse sequences for β-COP and δ-COP (mouse proteins were used for the EM structure) are 953 and 511 amino acids long, respectively, according to NCBI and Figure 5. However, the structure presented in Figure 1 as the "crystal structure of the β-δ-COP subcomplex" only contains β-COP 19-391 and δ-COP 1-175 from C. thermophilium (the COP proteins are highly conserved, so mixing organisms in the model should be fine). This is only 39% of β-COP and 34% of δ-COP. While this is not a problem in itself, the title of the Results section and the title of the figure legend should state that this is a "partial structure of β-δ-COP" or a "structure of the interaction between β-COP and δ-COP". In order to clearly convey which regions of the proteins are included in the structure, an inset should be added to Figure 1 that diagrams the domains of β-COP and δ-COP (similar to Figure 5), illustrating which parts are in the crystal structure and which parts are not.*

We have edited the title of results and figure legend, and have added an inset to Figure 1, as requested by the reviewer.

*2b) Figure 5 returns to the structure of β-δ-COP, this time looking at the subtomogram average to show the interaction with Arf1. Compared to the crystal structure in Figure 1, more (all?) of β-COP is displayed. However, the C-terminal half of δ-COP, corresponding to the MHD domain, is still absent. Later in the manuscript, it is briefly mentioned that the δ-COP MHD domain is part of the linkage between COPI triads (along with epsilon-COP and the C-terminus of α-COP; Figure 6—figure supplement 2, and crucially, only in the figure legend of Figure 2—figure supplement 2 is it mentioned that "δ-COP MHD is located in the linkage, and not in the COPI leaf, so it is not enclosed by the EM density". Finally, I understood why the MHD domain is not pictured in Figure 5, but this took far too much detective work. Thus: 1) the legend for Figure 2 should state which COP domains are not resolved within the 9A leaf structure due to their location in the linkage region outside of the triad, and 2) the legend of Figure 5 should reiterate that the MHC domain is not shown because it is outside the leaf structure.*

We have edited the legends as requested by the reviewer.

*3) Related to the above issue, additional labels should be added to many of the figures in order to make it easier to understand which components are being displayed in the images without having to continually return to the Figure 2 legend for the color code. If only specific domains are being shown instead of full proteins, this should also be labeled. This will make the results much easier to follow for readers who are not intimately familiar with the COPI structure. Specifically:*

*a) Color-coded labels on figure panels adjacent to the corresponding subunits:*

Figure 1: dark green "β-COP" (in addition to existing labels); Figure 3: red "appendage sandwich subdomain", purple: "appendage platform subdomain"; Figure 3: black "γ-COP appendage", light blue "β'-COP"; Figure 3: black "β-COP appendage", dark blue "α-COP"; Figure 4: light blue "β'-COP", dark blue "α-COP"; Figure 4: dark blue "α-COP N-terminus"; Figure 4: dark green "β-COP", orange "δ-COP", light green "γ-COP", yellow "zeta-COP", pink "Arf1"; Figure 5: dark green "β-COP", black "Longin domain" (in addition to "a", "b", "c"); Figure 7: red "ArfGAP2", pink "Arf1".

Changed as requested.

*b) The degree of rotation (e.g. 90°) should be added next to the rotation symbols in each of the Figure 2, Figure 3, Figure 4, Figure 5, Figure 2—figure supplement 2, Figure 7—figure supplement 1 Additional rotation symbols and degrees should be added to Figure 4.*

Changed as requested.

*c) In Figure 5, it would be very helpful to illustrate the position of the δ-COP "b" helix from the cryo-EM structure relative to Arf1-GDP in order to show how the N0 amphiphatic helix of Arf1-GDP blocks the binding site for δ-COP on Arf1. Since there is space, this could either be incorporated into the current image or be shown in a second image.*

The figure has been edited to illustrate this better.

*d) Figure 6 would be much more clear if the "a" and "b" helices of δ-COP were colored red and yellow, respectively, to be consistent with their coloration in Figure 1, Figure 5, and Figure 1—figure supplement 1. Upon doing so, the yellow spheres representing crosslinked residues should be recolored to match the red "b" helix (perhaps a dark orange would look good).*

Changed as requested.

*4) No PDB structures are listed for the atomic models that were fit into the EM density (α-COP, Arf1-GTP, etc.). While these structures may have been named in the previous Science paper, at least in the methods of the current manuscript, these structures should be listed and the appropriate papers should be cited.*

We have added the PDB IDs and citations to the Materials and methods section.

5) As in situ structures within cells are now possible, it should be made more explicitly clear in the abstract, introduction, and summary that the structure being presented is from an in vitro reconstituted system. Specifically:

Abstract: change "we determined the structure of the COPI coat assembled on membranes" to something like "we determined the structure of the COPI coat assembled on membranes in vitro".

Introduction: change "we recently determined the architecture of the COPI coat assembled on the vesicle membrane" to something like "we recently determined the architecture of the in vitro reconstituted COPI coat assembled on vesicle membranes".

*Summary: change "by combining the β-δ-COP crystal structure and the EM structure of the COPI coat on the vesicle membrane" to something like "by combining the β-δ-COP crystal structure and the* in vitro *EM structure of the COPI coat on the vesicle membrane".*

These changes have been made as requested.

*Reviewer #3:*

*1. δ-COP helices section. Does it make sense for helixC to be assigned to this position in the density? It seems located far away: what's the distance the δ-COP unstructured linker must span? And where does that imply the MHD sits? Why is helix c needed to help coordinate the MHD? This is very speculative, and the biochemical data don't seem to support the placement of helixC (below).*

According to the secondary structure predictions the linker between helices b and c is 60 amino acids long, so could span very large distances. The linker between helix c and the MHD is also long (18 amino acids) and is consistent with the observed position of the MHD. As clearly stated in the text, the assignment of helix c is speculative, and is based on the observed unoccupied density, and cross-linking data that place helix c close to β’. In our view the biochemical data do not contradict this speculative assignment (see below).

We have added a phrase: “*In this position, helix c could help to coordinate the positioning of the C-terminal δ-COP MHD, which could otherwise be located a long distance from the vesicle: there are ~100 amino acids between helix b and the MHD.*”

*2) The authors need to describe more clearly experimental details for pulldowns between β/δ-COP and Arf1 (GMPPNP) in the main text and figure labels (ie. strep-tagged sub-complexes on sepharose). Importantly, why don't we see two Coomassie bands in all lanes of Figure 6, corresponding to β- and δ-COP? And why haven't the authors shown pulldowns with both GDP- and GTP-locked forms of Arf1 if they say the interaction is nucleotide 'agnostic'?*

As requested, we have added further details on the pulldown experiments in the Figure 6 legend. The gel was divided into two pieces. The upper piece was used for coomassie staining and the lower piece was blotted for immunoreaction with Anti-Arf antibodies. The truncated δ-variants 1-137, 1-175 and 1-243 run into the lower part of the gel and therefore are not visible in the coomassie staining, whereas in all lanes the β-part is visible and therefore was used as loading control.

The physiological interaction of Arf1 with δ-COP is not nucleotide agnostic, rather it is nucleotide dependent. While we reference and explain the previous observation in the literature that N△17Arf1 can bind isolated δ in a nucleotide independent manner (Sun et al., 2007) we consider this physiologically irrelevant, and therefore didn´t follow it up in more detail here.

*Although the pulldowns demonstrate a biochemical interaction between purified β-/δ-COP and Arf1, such an interaction has been shown previously by Jonathan Goldberg. What's new here seems to be the importance of the linker region for the interaction. The authors state the main interaction partner of δ-COP helix b is Arf1, but their pulldown data indicate helix C is important (Figure 6). But then they place helixC very far away from Arf1 in the density (Figure 6). Why not do pulldowns with δ-COP linkers (of different lengths) and Arf1 in both nucleotide states to nail down this interaction? Is the affinity too low, even by Western blotting? Either way, the biochemistry suggests helixC is located much closer to Arf1. This story is interesting but is 'left hanging'.*

An interaction between δ-COP and Arf1 was reported in the literature by Sun et al., 2007 and we discussed those results in the manuscript. Goldberg and colleagues (Yu et al., 2012) reported an interaction between Arf1 and βδ-COP, which we have also addressed, though Yu et al., have only tested the Arf1-β interaction in their pull-down assays and not the Arf1-δ interaction. What we are able to do here is show an Arf1-δ interaction and describe the nature of the interaction.

Our biochemical data do not directly indicate that helix c is important, rather they indicate that a stretch of 60 amino acids including the long linker between helix b and c, as well as helix c, contributes to Arf1 binding. While other interpretations are possible, our preferred interpretation is that this stretch may help to stabilize binding of helix b to Arf1. We disagree that this data suggests helix c is much closer to Arf1, and have retained, unchanged, our cautious interpretation based on the cross-linking data: “Several cross-links (δ233-δ263, δ243-δ263, δ233-β’627 and δ241-β’627) ([Supplementary-material SD2-data]) suggest the approximate location of helix c (Figure 6—figure supplement 1), and based on these observations we speculate that density “5” is occupied by this α-helix (Figure 4 “5”).”

We agree that further experiments could be pursued to understand the role of helix c, but this will have to form part of a future study.

*3) The authors claim δ-COP helix b makes interactions with specific Arf1 regions (Switch I/Interswitch). I'm not convinced the resolution is sufficient to make these claims. And they have not shown us biochemical data to support the idea that δ-COP is agnostic to the nucleotide state of Arf1.*

This seems to be a misunderstanding – we do not think that the interaction of Arf1 with δ-COP is nucleotide agnostic, indeed our model addresses how δ-COP contributes to the nucleotide dependence of this interaction. The relevant paragraph of the manuscript has been edited in response to comments from the other reviewers, and we hope that this misunderstanding is now avoided.

The resolution of the structure is sufficient to define the orientation of Arf1 based on the position of α-helices, and also the position of helix b. In our opinion it is therefore entirely appropriate to interpret which regions of the Arf1 surface interact with the helix. We do not attempt to interpret which amino acids might be involved or not involved in interactions which would not be possible at this resolution.

*4) The authors propose that only the Arfs near γ-COP would be hydrolyzed by ArfGAP2? They suggest only these γArfs would be accessible for ArfGAP2 binding while βArfs would facilitate interaction with ArfGAP1. Is there evidence somewhere that supports the idea that different ArfGAPS bind different coatomer subunits (yeast two hybrid etc)? And is there any relation at all between one of these Arfs and the so-called 'backside' Arf1 seen in Hurley's open AP1 structure?*

Yes, beyond the structural evidence presented in our manuscript, other observations support this. We now refer to these experiments at two places: “*We were not able to resolve the C-terminal part of ArfGAP2, known to be involved in coatomer binding via interactions with the γ-COP appendage domain (Kliouchnikov et al., 2009; Watson et al., 2004).*” And “*This is consistent with yeast-two hybrid experiments which showed an interaction between the ArfGAP2 Glo3 and β’-COP and γ-COP but not with other coatomer subunits (Eugster et al., 2000).*”

No, there is no relation between the Arf1 interactions in the COPI coat and the “back side” interaction seen in the AP1 complex crystal. We have now commented on this: “*None of these contacts are similar to the previously described crystal contact between the α1 subunit of AP1 and the back side of βArf1 (Ren et al., 2013) suggesting that an equivalent interaction is not relevant within the COPI coat.*”

*5) The idea of non-equivalent Arfs could be tested biochemically by combining structure-based mutagenesis and their liposome budding assay to count budding vesicles per area. This would be an important confirmation of their structure-based hypothesis.*

The structure we present provides a wealth of data for designing structure-based mutagenesis experiments relating to multiple aspects of coat function and regulation. In our view, it is appropriate to publish the structure at this stage so that other researchers can take advantage of it.